# Automated measurement of anteroposterior diameter and foraminal widths in MRI images for lumbar spinal stenosis diagnosis

**Friska Natalia[1], Hira Meidia[1], Nunik Afriliana[1], Julio Christian Young[1], Reyhan Eddy Yunus[2], Mohammed Al-Jumaily[3], Ala Al-Kafri [4], Sud Sudirman [4]***

**1** Faculty of Engineering and Informatics, Universitas Multimedia Nusantara, Serpong, Indonesia, **2** Dr. Cipto Mangunkusumo Hospital, Jakarta, Indonesia, **3** Dr. Sulaiman Al-Habib Hospital, Dubai, UAE, **4** School of Computer Science and Mathematics, Liverpool John Moores University, Liverpool, United Kingdom

* s.sudirman@ljmu.ac.uk

**Data Availability Statement:** All relevant data required to replicate the study's results are publicly accessible via the following sources: Sudirman, Sud; Natalia, Friska (2020), "MATLAB source code for automated measurement of anteroposterior

## Abstract

Lumbar Spinal Stenosis causes low back pain through pressures exerted on the spinal nerves. This can be verified by measuring the anteroposterior diameter and foraminal widths of the patient's lumbar spine. Our goal is to develop a novel strategy for assessing the extent of Lumbar Spinal Stenosis by automatically calculating these distances from the patient's lumbar spine MRI. Our method starts with a semantic segmentation of T1- and T2-weighted composite axial MRI images using SegNet that partitions the image into six regions of interest. They consist of three main regions-of-interest, namely the Intervertebral Disc, Posterior Element, and Thecal Sac, and three auxiliary regions-of-interest that includes the Area between Anterior and Posterior elements. A novel contour evolution algorithm is then applied to improve the accuracy of the segmentation results along important region boundaries. Nine anatomical landmarks on the image are located by delineating the region boundaries found in the segmented image before the anteroposterior diameter and foraminal widths can be measured. The performance of the proposed algorithm was evaluated through a set of experiments on the Lumbar Spine MRI dataset containing MRI studies of 515 patients. These experiments compare the performance of our contour evolution algorithm with the Geodesic Active Contour and Chan-Vese methods over 22 different setups. We found that our method works best when our contour evolution algorithm is applied to improve the accuracy of both the label images used to train the SegNet model and the automatically segmented image. The average error of the calculated right and left foraminal distances relative to their expert-measured distances are 0.28 mm ($p$ = 0.92) and 0.29 mm ($p$ = 0.97), respectively. The average error of the calculated anteroposterior diameter relative to their expert-measured diameter is 0.90 mm ($p$ = 0.92). The method also achieves 96.7% agreement with an expert opinion on determining the severity of the Intervertebral Disc herniations.

diameter and foraminal widths in MRI images for lumbar spinal stenosis diagnosis", Mendeley Data, v1, http://dx.doi.org/10.17632/zwd3hgr6gg.1 Sudirman, Sud; Al Kafri, Ala; Natalia, Friska; Meidia, Hira; Afriliana, Nunik; Al-Rashdan, Wasfi; Bashtawi, Mohammad; Al-Jumaily, Mohammed (2019), "Lumbar Spine MRI Dataset", Mendeley Data, v2, http://dx.doi.org/10.17632/k57fr854j2.2 Sudirman, Sud; Al Kafri, Ala; Natalia, Friska; Meidia, Hira; Afriliana, Nunik; Al-Rashdan, Wasfi; Bashtawi, Mohammad; Al-Jumaily, Mohammed (2019), "Label Image Ground Truth Data for Lumbar Spine MRI Dataset", Mendeley Data, v2, http://dx.doi.org/10.17632/zbf6b4pttk.2 Sudirman, Sud; Al Kafri, Ala; Natalia, Friska; Meidia, Hira; Afriliana, Nunik (2019), "MATLAB source code for developing Ground Truth Dataset, Semantic Segmentation, and Evaluation for the Lumbar Spine MRI Dataset", Mendeley Data, v2, http://dx.doi.org/10.17632/8cp2cp7km8.2 Sudirman, Sud; Al Kafri, Ala; Natalia, Friska; Meidia, Hira; Afriliana, Nunik; Al-Rashdan, Wasfi; Bashtawi, Mohammad; Al-Jumaily, Mohammed (2019), "Radiologists Notes for Lumbar Spine MRI Dataset", Mendeley Data, v2, http://dx.doi.org/10.17632/s6bgczr8s2.2.

**Funding:** FN Grant Number: 031/AKM/MONOPNT/2019 Funder: Indonesian Ministry of Research, Technology and Higher Education https://international.ristekdikti.go.id/ Statement: The funder had no role in study design, data collection and analysis, decision to publish, or preparation of the manuscript.

**Competing interests:** The authors have declared that no competing interests exist.

# 1. Introduction

Lumbar Spinal Stenosis (LSS) causes low back pain through pressures exerted on the spinal nerve and could result in *sciatica* which symptoms include radicular pain, atypical leg pain, and neurogenic claudication [1]. A review of the literature shows some loose consensus that categorizes low back pains that persist for more than twelve weeks as Chronic Lower Back Pain (CLBP). An economic impact study on CLBP [2] concluded that the success chance of a patient's rehabilitation depends largely on early identification of the cause of the back pain.

Unfortunately, due to a heavy demand for neuroradiologists and specialists, early identification may not always be possible. The overall time for a diagnosis could take many weeks to complete since it may include referral and waiting time for a specialist doctor and the time for a medical scan and the subsequent analysis. In England for example, the NHS reported a significant and increasing number of cases where the wait time for diagnostic radiology exceeds its maximum target of thirteen weeks [3]. This problem is expected to deteriorate since the number of scans always increases historically. The Royal College of Radiologists reported [4] that three-quarters of UK medical imaging departments do not have sufficient radiology consultants to deliver safe and effective patient care, and subsequently, more money is being spent to pay outsourcing, overtime, and locums to cover radiologist work every year. This problem is exacerbated by the fact that there is a 12.3% average annual growth in demand for radiographical imaging including Magnetic Resonance Imaging (MRI) and Computed Tomography (CT) scans since 1995 [5]. This rationalizes the need for a new approach to increase the efficiency and effectiveness of the imaging diagnostic process. Hence, in this paper, we propose a methodology that could help improve the situation, particularly in LSS diagnosis through automatic measurement of anteroposterior (AP) diameter and the left and right foraminal widths on MRI images.

# 2. Background and related work

The size of the spinal cord is generally considered as an important factor in the symptomatology of LSS [6]. Together with our brain, the spinal cord makes up our central nervous system. It is a cylindrical structure that starts at the end of the brain stem and continues down the spine and ends where *cauda equina* starts. The spinal cord and cauda equina contain a bundle of nerves that are enclosed by a membranous sheath containing the *thecal sac* (TS). Nerve signals from the brain to other parts of the body and vice versa are transmitted via motor roots and sensory roots that are contained in the TS. These roots exit the spinal cord and cauda equina through the left and right openings between vertebrae as spinal nerves and caudal nerve roots, respectively. A narrowing of the *osteoligamentus* vertebral canal and/or the intervertebral foramina could cause direct or indirect compression of nerve bundles in the TS and their nerve roots [7]. Fig 1 exemplifies such compression of the nerve bundles and nerve roots, creating a central and lateral foraminal stenosis, around an L5-S1 Intervertebral Disc (IVD) viewed from the axial perspective.

Most LSS occurs in the last three IVDs namely L3-L4, L4-L5, and L5-S1. A neuroradiologist almost always starts his or her inspection of these IVDs in the mid-sagittal view since it can provide a general overview of the lumbar spine. This is reflected also by the popularity of automated inspection of sagittal MRI to detect and localize IVD herniations [8–12]. However, a more accurate assessment of the actual location and extent of the stenosis can only be obtained through inspection of each IVD in axial view [13, 14]. An axial slice that cuts closest to the half-height of each IVD is generally considered as the best slice to use when the neuroradiologist inspects the disc. He or she then manually locates several regions of interest (ROIs) including three important ones namely, the IVD, the posterior element (PE), and the TS with the

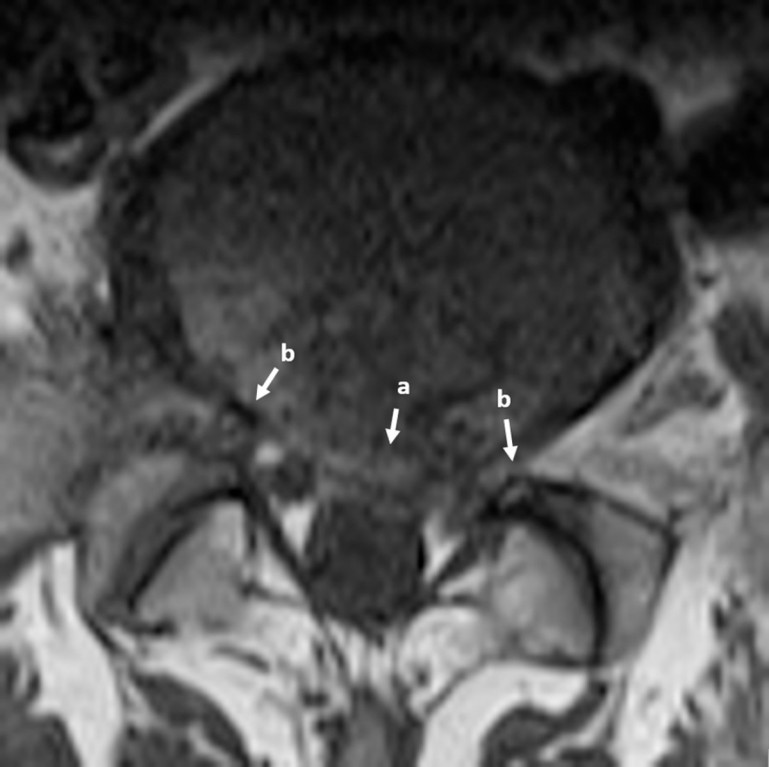

**Fig 1.** Axial view MRI of L5-S1 IVD showing a) central stenosis and b) left and right lateral foraminal stenosis, both of which impinging upon the nerve bundle contained in the thecal sac and the nerve roots exiting the spine.

view of using them to calculate three measurements namely the AP diameter and the left and right foraminal widths. The process may take several minutes since it often requires the neuro-radiologist to scan the entire image and to zoom in and out of specific parts of the image. Computer assistance is necessary to reduce the occurrence of human errors and to speed up the process. This assistance may include an accurate delineation of the bottom boundary of the IVD region and the top boundary of the PE region and automatic measurement of the AP diameter and the left and right foraminal widths. These boundaries and the three measurements are illustrated in Fig 2.

There are several methods proposed in the literature to localize LSS. Zhang et al. [15] proposed using a weakly-supervised classifier, which had been trained using pathological labels, to identify vertebrae regions in axial MRI images to localize LSS. A stochastic objective function, optimized using the Regularized Dual Averaging algorithm, is used to perform the learning process. The paper claims that the weakly-supervised classifier performs better than the strongly-supervised alternatives.

SpineNet [16] is a convolutional neural network (CNN) framework that localizes and predicts radiological grading of pathologies on lumbar spine MRI images using IVD volumes as its inputs. SpineNet predicts several gradings at the same time and can be trained using a multi-task loss function without needing segmentation level annotation. It is tested on a dataset that contains T2-weighted sagittal MRIs acquired from multiple machines. The paper claimed a "near-human" performance in grading IVD degeneration using Pfirrmann grading, disc narrowing, upper/lower endplate defects, upper/lower marrow changes, spondylolisthesis, and central canal stenosis.

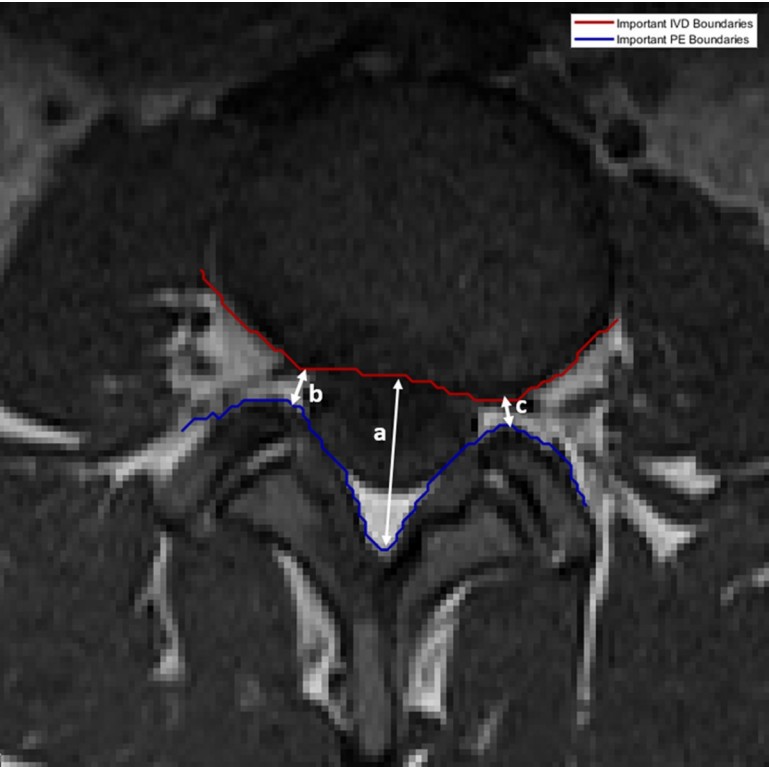

**Fig 2. The three measurements that are necessary to determine the extent of LSS.** The a) AP diameter is used to determine how much pressure exerted on the spinal cord, b) left and c) right foraminal widths which are used to determine the pressure exerted on left and right spinal nerve roots, respectively. The red and blue lines delineate the important boundaries of IVD and PE regions, respectively.

DeepSpine [17] was proposed to fully-automate LSS grading by leveraging expert's information stored in large-scale archived free-text radiology reports and MRI images. The method used a natural-language-processing scheme to extract information from the reports as ground-truth labels for the various types and grades of LSS. It applied a U-Net architecture combined with a spine-curve fitting method to segment and localize vertebrae in the axial and sagittal MRI images. A multi-class CNN, trained using both the extracted label and segmented images, is then used to perform central canal and foraminal stenosis grading. The paper reported a significant improvement to the overall classification accuracy compared to [15, 16] when tested on a proprietary dataset containing 22,796 disc-levels MRI images of 4,075 patients.

Accurate delineation of region boundaries in an image requires image segmentation, a process that partitions an image into multiple segments, where all pixels belonging to each segment having a set of common characteristics. There are several approaches in the literature that segment lumbar spine MRI images including one for spinal canal detection [8], and for vertebrae and IVD detection [10, 12]. In the more general field of image processing, active contour model is a popular class of image segmentation and boundary delineation method due to its ability to fit a curve to an object boundary. It works by, given an initial boundary estimate, iteratively expanding or contracting the estimate to minimize an energy function. Several active contour models have been used to segment and delineate boundaries in different types of medical images [18–20].

We have, however, previously shown [21, 22] two drawbacks in existing active contour models. Firstly, existing approaches cannot apply contour evolution to only specific parts of

the contour. Having the ability to decide which part of the contour to evolve and which feature to use, would allow for a more adaptive contour evolution. Secondly, the inclusion of all control parameters when calculating the energy function minimization makes it harder to find the right combination that yields the best results. The effect of adjusting one control parameter may counter the effect of adjusting the others which results in finding the right combination of parameter values involves trial-and-error and can be very tricky and case dependent. In the next section, we will describe the approach we have taken in using our contour evolution method to improve the result of automated image segmentation, and in turn, allows for an accurate location of region boundaries and landmarks in the image to measure the AP diameter and foraminal widths.

## 3. Material and method

### 3.1 Lumbar spine MRI dataset

One of the many issues in the methodologies proposed in the literature is that they are tested on a proprietary dataset which makes it difficult to verify and reproduce. In this study, we use our open-access Lumbar Spine MRI dataset [23] collected from 515 patients with symptomatic back pain. We can confirm that all procedures performed in this study are in accordance with the ethical standards of both the United Kingdom and the Kingdom of Jordan and comply with the 1964 Helsinki declaration and its later amendments. The approval was granted by the Medical Ethical Committee of Irbid Speciality Hospital in Jordan.

Each MRI study is annotated by expert radiologists with information on the condition and observed characteristics of the lumbar spine, and presence of diseases which include central/foraminal stenosis, spondylolisthesis, *ligamentum flavum* hypertrophy, annular tears, endplate defects (Modic type) and degeneration, facet joint defects, scoliosis, TS compressing, IVD bulges and bone marrow disease.

### 3.2. The proposed methodology

Our methodology to measure the AP diameter and foraminal widths starts with selecting an axial slice that cuts closest to the mid-height of each of the last three IVDs [14] from the T1 and T2 sequences of each patient's MRI study. We then apply image registration to each T1- and T2-weighted image pair to ensure that every pixel at the same location in both images corresponds to the same point in an organ or tissue. This is performed by finding the minimum difference between the fixed T1-weighted image and a set of transformed T2-weighted images calculated over a search-space of affine transforms. The minimum and maximum limits of the radius of the search-space are set to 1.5e-6 and 13e-3, respectively. The search is carried out up to 300 iterations with a parameter growth factor of 1.05. To counter the effect of high-frequency noise and low-frequency inhomogeneity field on both T1 and T2 modalities, a parametric bias field estimation is applied before being corrected using the PABIC method [24]. A search optimization algorithm called (1+1)-Evolutionary Strategy is employed by locally adjusting the search direction and step size while at the same time provides a mechanism to step out of non-optimal local minima. After the registration process completes, a composite 3-channel image is created using the fixed T1-weighted and transformed T2-weighted images for the first and second channels. The third channel is set to the Manhattan distance of the two images.

The resulting 1545 composite images are then randomly split into a training and a test set, containing 80% and 20% of the overall dataset, respectively. The images in the training set are used to train a SegNet model [25]. The training process starts by developing label images by manually annotating several ROIs in each composite image. For each label image, two specific

region boundaries, namely important IVD boundary and important PE boundary, are located. These boundaries are important because they are used to locate certain points or landmarks from which the AP diameter and foraminal widths can be measured. Since the accuracy of locating these boundaries directly affect the accuracy of the AP diameter and foraminal widths measurements, we decide to improve the accuracy of the label images along these boundaries using a contour evolution technique [21]. These improved label images and their corresponding composite images are then used to train the SegNet model. This model is subsequently used to automatically segment the composite images in the test dataset. The contour evolution technique is then applied to the resulting segmented images to further improve the region boundaries before the important IVD boundary and the important PE boundary are located. From these boundaries, we locate several important points that are used to measure the AP diameter and foraminal widths [26].

We have previously shown the applicability of the SegNet architecture in segmenting axial MRI images using unmodified label images [27]. We will use the previously reported results as a baseline to measure the improvement in the segmentation accuracy along the important boundaries using the proposed method. An overview of the proposed methodology is elucidated in a flowchart shown in Fig 3. The remainder of this section will elaborate on three main elements of our methodology.

**3.2.1. Segmentation of regions of interest.** The first main step in our approach is to automate the process of identifying the three important ROIs through image segmentation of axial MRI images using a SegNet model. The input to the SegNet model is a composite image containing the T1-weighted image, the registered T2-weighted image, and their Manhattan distance. The model training process is guided using manually developed label images containing the three important regions stated previously (IVD, PE, and TS) plus three other regions. The first one is the *Area between Anterior and Posterior* (AAP) vertebrae elements. This region represents the space that extends from the cervical spine down to the lumbar spine between the front and back vertebrae elements. In the axial view, this region is the region between the IVD and PE. The second region contains a set of pixels where T1-weighted pixels have no correspondence with any of the transformed T2-weighted pixels. We refer to this as the 'Unregistered' region. Any other pixels that do not belong to any of the above five regions are labeled as 'Other'. Each of these regions represents a class for classification purpose and as a reference for the remainder of this paper, they are identified with their unique identification numbers as 1 = Unregistered, 2 = IVD, 3 = PE, 4 = TS, 5 = AAP, and 6 = Other. An example of such a label image and its corresponding composite input image are shown in Fig 4.

To help formulate the definition of the important IVD and PE boundaries, the set descriptor of the points along these boundaries is provided as follows. Important IVD boundary, denoted as $A_2 \in \Re^2$, is defined as a set of points that are adjacent to the IVD region and the

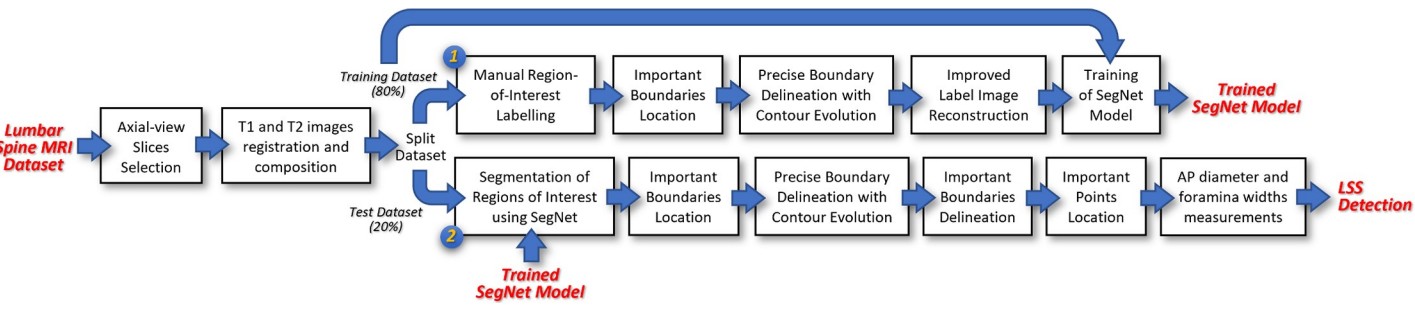

**Fig 3. An overview of the overall methodology.**

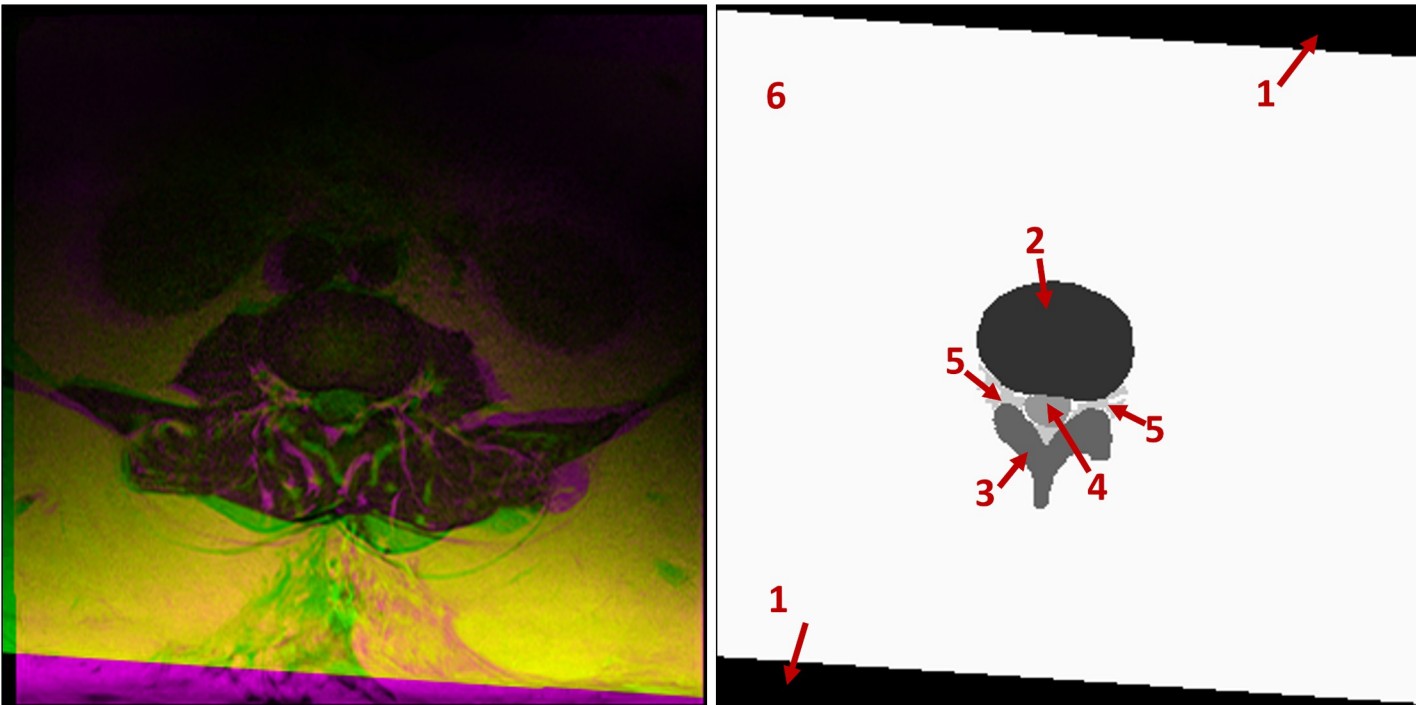

**Fig 4. An example of the T1- and T2-weighted composite input image (left) and the label image used to train the SegNet model (right).** The regions in the label image are 1) Unregistered, 2) IVD, 3) PE, 4) TS, 5) AAP and 6) Other.

AAP, TS, or PE regions. Equally, important PE boundary, denoted as $A_3 \in \Re^2$, is defined as a set of points bordering between the PE region and the AAP, TS, or IVD regions. They are defined as a union of each boundary set that makes them:

$$A_2 = E_{23} \cup E_{24} \cup E_{25}$$

$$A_3 = E_{32} \cup E_{34} \cup E_{35}$$

Where $E_{ab} \in \Re^2$ is a set of 2D coordinate of the boundary points between region $a$ and region $b$.

**3.2.2. Precise boundary delineation with contour evolution.**    The accuracy of the label images used to develop a classifier model plays a significant part in determining the accuracy of the model. Simply put, training a classifier model with inaccurate label images would produce an inferior model. However, the process of manually annotating label images is very laborious and error-prone. Even when the process is done by a skilled labeler, there is still a high level of probabilities of inaccuracies, especially along region boundaries. To illustrate this point, we show in Fig 5 an example of the region boundaries of a manually created label image and its corresponding automatically segmented image using the SegNet model we developed previously in [27]. Fig 5A clearly shows the challenges in manual segmentation, especially around the aforementioned important boundary areas. The figure shows numerous holes at various locations along the important boundaries in the manually created label image. Although the automatically segmented image shown in Fig 5B is relatively free of these holes, it still inherits the inaccuracy from the training label images. We aim to resolve this problem by improving the segmentation accuracy along these boundaries through an application of our contour evolution method.

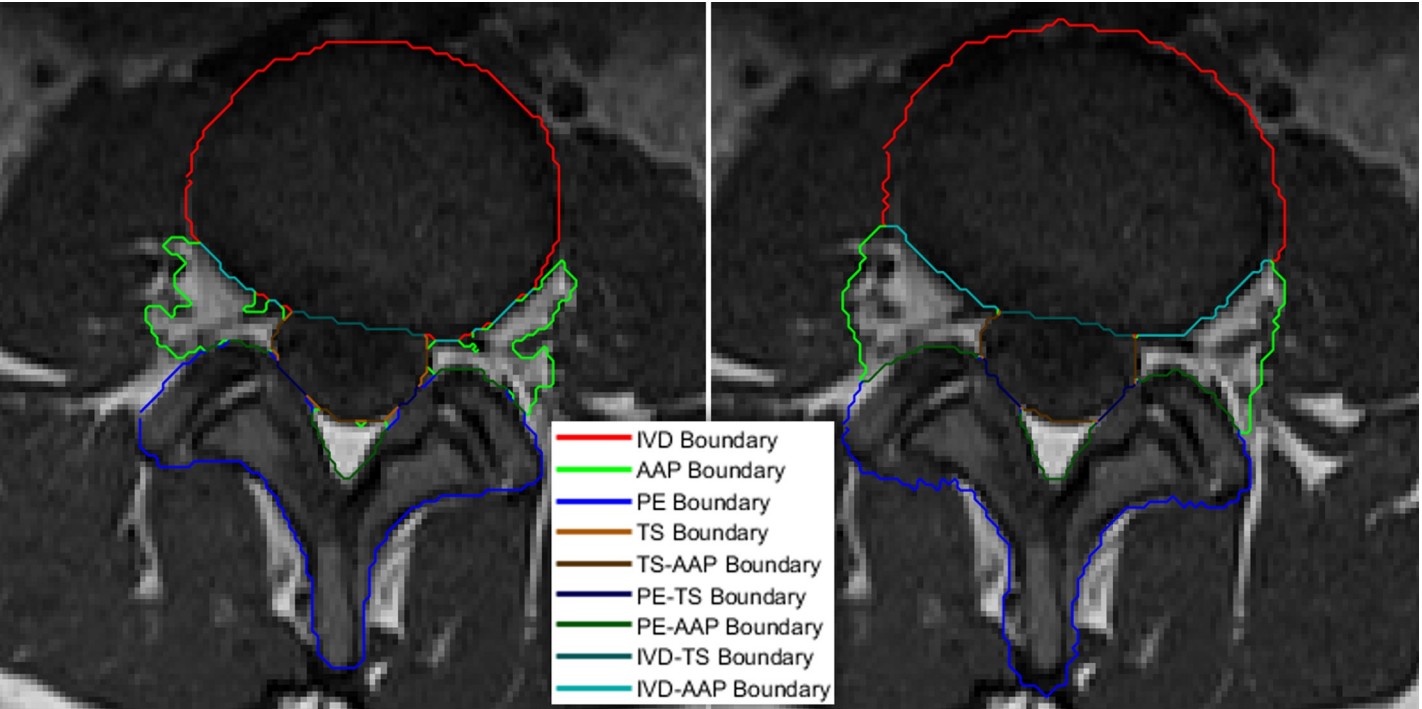

**Fig 5. A color-coded boundary of a manually labeled lumbar spine MRI image (left) and the automatic segmentation result (right).** The left image clearly shows ambiguities and inaccuracies, particularly along the important boundary locations.

Our approach to contour evolution to obtain more precise boundary delineation uses two structures, namely the *Modified Boundary Grid* and *Sparse Boundary Representation*, originally proposed in [28] to represent the segmented image and the contour information, respectively. The description of these two structures and the algorithms to evolve them are described as follows:

The most straightforward way to represent the regions in an image is by using a matrix of labels with the same size as the original image. As an example, an instance where a 4×4 image that is segmented into three regions is illustrated in Fig 6. The figure depicts the region's boundaries as red lines between pixels that have different labels.

Boundary locations are naturally sub-pixels because they exist between two adjacent pixels. They can be stored as a *Boundary Grid* which is a matrix whose size is twice that of the label image minus one. A version of the Boundary Grid that was proposed in [28] using the label image is shown in Fig 6 (bottom).

In the above example, the cells with boundary information are shaded in red and their cell values signify the normalized boundary strength. It is worth noting that not all cells in a Boundary Grid can be boundary cells and in general only cells that have either 1) an even row number and an odd column number or 2) an even column number and an odd row number, can be boundary cells. Those cells that do not belong to either category cannot be boundary cells and are marked with a red cross. Furthermore, boundary cells in category (1) mark only horizontal boundaries whereas those in category (2) mark only vertical boundaries. Any of the cells that meet the criteria but are not boundary cells have a cell value of 0. The rest of the cells contain the corresponding input image pixel values.

Our modification to this approach is twofold. First, instead of storing the input image pixel values, we replace them with the label pixel value. The rationale for this change is, since we are

|   | 1 | 2 | 3 | 4 |
|---|---|---|---|---|
| 1 | 1 | 1 | 3 | 51 |
| 2 | 2 | 1 | 50 | 49 |
| 3 | 244 | 242 | 244 | 50 |
| 4 | 240 | 241 | 242 | 241 |

|   | 1 | 2 | 3 | 4 |
|---|---|---|---|---|
| 1 | $x$ | $x$ | $x$ | $y$ |
| 2 | $x$ | $x$ | $y$ | $y$ |
| 3 | $z$ | $z$ | $z$ | $y$ |
| 4 | $z$ | $z$ | $z$ | $z$ |

|   | 1 | 2 | 3 | 4 | 5 | 6 | 7 |
|---|---|---|---|---|---|---|---|
| 1 | 1 | 0 | 1 | 0 | 3 | 1 | 51 |
| 2 | 0 | X | 0 | X | 1 | X | 0 |
| 3 | 2 | 0 | 1 | 1 | 50 | 0 | 49 |
| 4 | 1 | X | 1 | X | 1 | X | 0 |
| 5 | 244 | 0 | 242 | 0 | 244 | 1 | 50 |
| 6 | 0 | X | 0 | X | 0 | X | 1 |
| 7 | 240 | 0 | 241 | 0 | 242 | 0 | 241 |

**Fig 6.** An example of a 4x4 image (top left) that is segmented into a label image containing three regions with red lines marking the boundaries between them (top right) and its Boundary Grid (bottom).

using this structure to evolve the boundary, we will need to be able to also modify the relevant pixel's label. The resulting label image can be simply reproduced by downsampling the Boundary Grid at every odd pixel location. Our second modification is to use the boundary cells, including the red-shaded boundary cells, to store image feature values instead of edge strengths. This image feature will be used as one of the factors to evolve the boundary. The type of image feature chosen is a design decision and is important to the accuracy and suitability of the algorithm to any given problem. In this paper, we use the image horizontal and vertical gradients as the chosen feature to use.

We refer to this modification as the *Modified Boundary Grid*, denoted as $B'$. The image function of $B'$ is given as:

$$B' : \begin{cases} \Omega \to \Gamma | (\Omega_1 \bmod 2) \neq 0 \wedge (\Omega_2 \bmod 2) \neq 0 \\ \Omega \to \Re | (\Omega_1 \bmod 2) = 0 \oplus (\Omega_2 \bmod 2) = 0 \\ \Omega \to \varnothing | \text{otherwise} \end{cases}$$

where $\Omega_i$ is the $i^{th}$ element of $\Omega$ and $\Gamma = \{1, \ldots, N\}$ is region label where $N$ is the maximum number of regions in the image.

The algorithm to construct our $B'$ is as follows: Let $I: \Omega \to \Re$ be the input image function and $\chi: \Omega \to \Gamma$ be its associated label image function, both with domain $\Omega \subset \Re^2$. At locations where $(\Omega_1 \bmod 2) \neq 0 \wedge (\Omega_2 \bmod 2) \neq 0$, $B'$ contains the label information as prescribed in $\chi$. At locations $(\Omega_1 \bmod 2) = 0 \wedge (\Omega_2 \bmod 2) \neq 0$, $B'$ contains vertical gradients and at locations $(\Omega_1 \bmod 2) \neq 0 \wedge (\Omega_2 \bmod 2) = 0$, $B'$ contains horizontal gradients.

In practice, the gradient values are calculated using the convolution of the input image $I$ with a Gaussian kernel $G_\sigma$ of width $\sigma$ to minimize high-frequency elements in the derivative operation. Hence the horizontal and vertical gradient image functions $g_h: \Omega \to \Re$ and $g_v: \Omega \to \Re$ are defined as $g_h = \nabla_h (G_\sigma * I)$ and $g_v = \nabla_v (G_\sigma * I)$, respectively. The sizes of the horizontal and vertical gradient images are $H \times (W-1)$ and $(H-1) \times W$, respectively. To illustrate the process to construct our $B'$, we show the horizontal $g_h$ and vertical $g_v$ gradient images of the input images we used previously in the top left and the top right of Fig 7, respectively. The resulting modified Boundary Grid $B'$ is shown at the bottom of Fig 7. Note the boundary cells now contain values of the image gradient features.

The Modified Boundary Grid structure is, however, neither efficient nor compact if we want to use it in an iteration to search for the coordinates of a specific region's boundary. To compensate for this shortcoming, we also use a *Sparse Boundary Representation*, denoted as $B_S$, as a look-up table to index pairs of neighboring regions. This is a much more compact and efficient representation of the boundary information because any search or information retrieval operations that are carried out in this representation are carried out faster due to them being dependent only on the number of boundary pixels rather than the number of pixels. We can construct $B_S$ from $B'$ by parsing the latter once, to gather all edge coordinates for each neighboring region pairs. Table 1 shows the Sparse Boundary Representation of the Modified Boundary Grid example we used earlier. Note that matrix subscripts are expressed as row and column order.

Contour evolution using our method is achieved by applying the SubPixelBoundaryEvolution algorithm on the Modified Boundary Grid and the Sparse Boundary Representation structures. The algorithm iteratively interpolates the Modified Boundary Grid and makes a call to

**Fig 7.** The horizontal gradient image (top left) and the vertical gradient image (top right) and the Modified Boundary Grid (bottom) of the 4x4 input and label images shown at the top of Fig 6.

another algorithm namely EvolveBoundaries that performs the boundary evolution. An overview of the process is illustrated in Fig 8.

The description of the method and algorithms are as follows. Let $S$ be a set containing the matrix subscript pairs of all the boundary points that we want to evolve using image gradient features. We construct $S$ directly by querying $B_S$. For each element in $S$, we find its location in $B'$ and, based on the values of the coordinate, we can ascertain the type of edge it is. A horizontal edge is evolved horizontally by evaluating the feature vectors in the horizontal direction. Likewise, vertical edges are evolved vertically. We need to decide the value of the search width, $w$, (in pixel) of the evolution, where $w$ is a positive integer number. The value of $w$ affects the speed and accuracy of the evolution and typically we want to use a small number. Large $w$ value may result in a label that is very different from the original estimate. The evolution of the boundary contour will be based directly on $S$ and is done by altering the label contents of $B'$ as described in Algorithm 1 in S1 File.

Our new contour is then obtained directly by parsing the evolved $B'$ while generating the new $B_S$. The algorithm is executed iteratively until the number of pixels in the Boundary Grid that change from one iteration to the next, averaged over $n$ number of iterations, converges below a specified threshold $\tau$.

To further improve the precision of our method, we upsample our input and label images at the end of each complete evolution. We can repeat this for $k$ number of times, where $k \in \mathbb{N}$, to increase the boundary precision to the nearest $2^{-(k+1)}$ of a pixel as described in Algorithm 2 in S1 File.

We design our method to allow the decoupling of the different parameters of the contour evolution process. To this effect, we apply an adaptive curve smoothing function at the end of the above contour evolution process. The rationale for this is to allow us to adjust the tightness of the curve to the feature and only concern about smoothing the results afterward. The curve smoothing process is image-feature dependent and implemented using variable width moving average method. Let $A = \langle p_n \mid n \in \mathbb{N} \text{ and } n < \text{card}(A) \rangle$ where $p_n \in \Re^2$ is a sorted sequence of set $S$. We define the sort operation such that $\|p_n - p_{n-1}\|$ for $1 \leq n \leq \text{card}(A)$ is minimized. We then apply a moving average on $A$ at every point $p$ along its curvature with variable half-width $w_p \in \mathbb{N}$ such that:

$$p_i = \frac{\sum_{j=-w_p}^{w_p} p_{i+j}}{2 \times w_p + 1}$$

For $\forall p$ that meets the $w_p < p < \text{card}(A) - w_p$ requirement. The value of $w_p$ is set between $w_{\min}$ and $w_{\max}$ and tied to the value of the image feature at the location of the boundary points. In our experiment, we use the second derivative of the image function $I''$ as the image feature and set $w_p$ to $w_{\min}$ when the $I''$ is at its lowest value, to $w_{\max}$ when the $I''$ is at its highest value and linearly interpolated and rounded to the nearest integer when in between.

**Table 1. Sparse boundary representation.**

| Neighbors | Edge Matrix Subscripts (S) |
|-----------|----------------------------|
| x,y | (1,6) (2,5) (3,4) |
| x,z | (4,1) (4,3) |
| y,z | (4,5) (5,6) (6,7) |

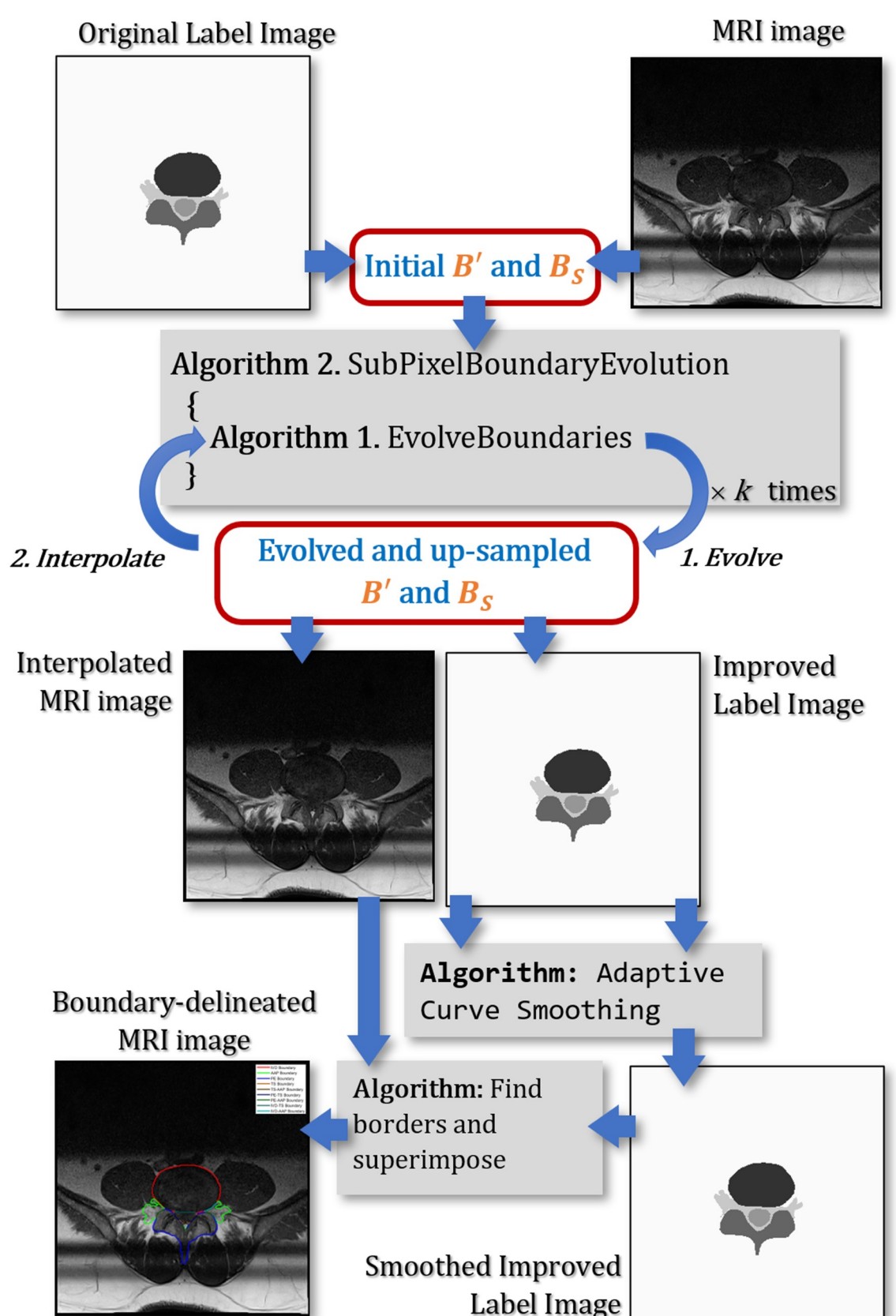

**Fig 8. An overview of the proposed contour evolution process.**

The contour evolution technique is applied to the manually created label images in the training set before they are used to train the SegNet model. They are also applied to the automatically segmented images in the test set before important points or landmarks are located on them.

**3.2.3. Important points location.** The procedure to measure the left and right foraminal widths and the AP diameter requires identifying and locating a set of important points on the MRI image. These points are the $T_L$, $T_R$, $B_L$, $B_R$, $B_M$, and $D_C$, which locations are shown in Fig 9. Points $T_L$ and $B_L$ are the top and bottom endpoints of the shortest line segment connecting the upper and lower part of the left foramina, respectively. Similarly, points $T_R$ and $B_R$ are their equivalent on the right foramina. Point $B_M$ is the lowest point of the top boundary of the PE region between $B_L$ and $B_R$ points, whereas $D_C$ is the centroid of the IVD region.

The line segment connecting $B_M$ and $D_C$, denoted as $B_M$-$D_C$, approximates a line that bisects the IVD region. This bisecting line is important and will be used to aid the measurement of the AP diameter since any compression on the spinal cord can be observed in the region around this line. The AP diameter can be estimated as the distance between $B_M$ and $q'$, a point that lies on the $B_M$-$D_C$ line segment. Point $q'$ is the projection of point $q$ on to the $B_M$-$D_C$ line. Point $q$ is a point that lies on the important IVD boundary (marked in red) between $T_L$ and $T_R$. The location of $q$, hence the location of $q'$, relative to the $T_L$-$T_R$ line segment can indicate the presence of IVD herniation.

Yates et. al. investigated the influence of the IVD shape and size on the pathway of herniation [29]. Through an *in-vitro* examination of cadavers, they concluded that the shape of the IVD influences stress distributions within it. It is also reported that the mean shape of a healthy IVD in a range of horizontal cross-section MRI images resembles a kidney-shaped [30]. Based on these, we postulate that the important boundary of a healthy IVD should be above the

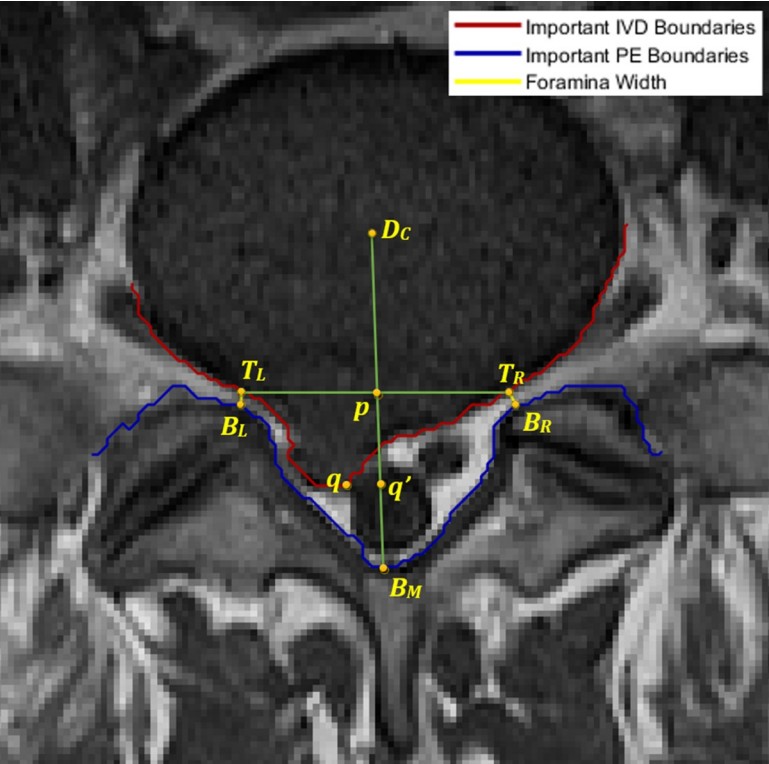

**Fig 9. The location of nine important points for determining AP diameter and foraminal widths.**

$T_L$-$T_R$ line segment and the distance between them is maximum at around its center. If the IVD is expected to be healthy, point $q$ is determined as the furthest point on the red boundary line to both the $T_L$-$T_R$ line segment and the $B_M$ point. On the other hand, if the red boundary line is below the $T_L$-$T_R$ line segment, an IVD herniation exists. The neurologist can estimate the extent of the IVD herniation, if it exists, by observing how far the red boundary line is relative to the $T_L$-$T_R$ line segment. The further down the red line is, the more severe the herniation. The location of point $q$, in this case, is determined as the furthest point on the red line to the $T_L$-$T_R$ line segment that is also closest to the $B_M$ point. Using the derivation we provided in [26], the point $q$ can be found using:

$$q = \arg\max_{z \in M_{\text{IVD}}}(|z \cdot \overline{T_L T_R}| - d(z, B_M))$$

where $M_{\text{IVD}}$ is the set of points on the important IVD boundary bounded by $T_L$ and $T_R$, and where $d()$ and $\cdot$ are the Euclidean distance operator and the dot product operator, respectively. The state of disc herniation can be categorized using the ratio $r = d_{q'} / d_p$ where $d_{q'} = d(B_M, q')$ and $d_p = d(B_M, p)$, and point $p$ is the intersection of $T_L$-$T_R$ and $B_M$-$D_C$ line segments. A healthy kidney-shaped IVD would result in $r \geq 1.1$, which means point $q'$ is sufficiently above the $T_L$-$T_R$ line segment. A herniated IVD, on the other hand, would result in smaller values of $r$. We further classify this case into two sub-categories, namely minor and severe herniations. A severe IVD herniation is detected when $d_{q'}$ is much shorter than $d_p$ and minor herniation is when $d_{q'}$ is slightly shorter or about the same length as $d_p$. We use a threshold value of 0.8 to differentiate the two instances, i.e., severe herniation when $r \leq 0.8$ and minor herniation when $0.8 < r < 1.1$.

## 4. Experiment setup, result analysis, and discussion

We conducted two types of experiments to demonstrate the workflow and to evaluate the performance of the proposed method. The first experiment is designed to show the result of the segmentation and the improvement achieved by the contour evolution technique. The second experiment is designed to show the performance of the overall approach in measuring the AP diameter and foraminal widths and classifying the IVD herniation.

### 4.1. Segmentation and boundary delineation

**4.1.1. Experiment setup and performance metrics.** We adopted the transfer learning approach [31] when training the SegNet model instead of developing the model from scratch since we have shown previously that the former approach produces significantly better segmentation than the latter approach [27]. We did this by using a SegNet model with pre-trained VGG16 coefficients on the ImageNet database [32]. The classification layer was replaced with a new randomly initialized fully connected neural network before the whole network is retrained using the training dataset.

During the training process, the SegNet model's weights and biases are updated using the popular Stochastic Gradient Descent with Momentum algorithm [33] with a learning rate of 0.001. The size of the training batch is 40 and the maximum number of epochs is 100. The input images need to be resized to 360×480 to match the input size requirement of the VGG16 network. Furthermore, to balance the importance of all classes due to class population imbalance, we apply class weighting [34]. To make sure that small-sized classes, such as the TS and AAP, are not underrepresented in our training data we set the class weighting to be inversely proportional to the class population. The segmented images produced by the trained model are then resized back to 320×320 before being used in the rest of the workflow.

The segmentation performance along the two important boundaries is measured using a semantic contour-based metric namely the BF-score [35], denoted as $F_c$, of every class $c$. The metric is calculated as:

$$F_c = \frac{2 \times P_c \times R_c}{P_c + R_c}$$

Where $P_c$ and $R_c$ are the precision and recall [36] for each class $c$ at a distance threshold $d_T$, respectively, and calculated as:

$$P_c = \frac{1}{|B_{pc}|} \sum_{z \in B_{pc}} [d(z, B_{gc}) < d_T]$$

$$R_c = \frac{1}{|B_{gc}|} \sum_{z \in B_{gc}} [d(z, B_{pc}) < d_T]$$

Where $B_{pc}$ and $B_{gc}$ are the sets containing the coordinates of the contour of the region of class $c$ from the predicted and ground-truth segmentation images, respectively. The function $d(z,B)$ denotes the shortest Euclidian distance between point $z$ and all the points in set $B$, and $d_T$ denotes the distance error tolerance.

**4.1.2. Experiment results and analysis.** The experiment was implemented using MATLAB on a Windows 10 PC with i7-7700 CPU @ 3.60GHz, 64 GB RAM, and two NVIDIA Titan X GPUs. To allow for comparative analysis, two other active contour methods were also implemented. They are geodesic active contour (GAC) by Caselles [37] and the active contour without edges by Chan and Vese (CV) [38]. We experimented with several combinations of $\alpha$, $\beta$, and $\gamma$ parameter values when implementing the GAC method to produce an acceptable compromise between accuracy and smoothness, and we found that $\alpha = 1$, $\beta = 160$, and $\gamma = 500$ are the best combination to use. We employed a convergence test at the end of each iteration of both methods to terminate the boundary evolution. The test checks if the number of pixel changes between successive iterations falls under a threshold. The threshold value is dependent on the number of pixels in the image. We also applied a morphological closing on the input label image prior to applying each technique to remove any small 1×1-sized holes and gaps in the image.

As a baseline, we use the BF-Score calculated between unmodified manually labeled images vs. unmodified predicted label images produced by a SegNet model trained using the unmodified manually labeled images. This is shown in the first row of Table 2. The result shows that only 68% of the important IVD boundary points and 79% of the important PE boundary points of the segmented images are correctly located within one pixel of their actual locations.

**Table 2. Averaged BF-score of important IVD and PE boundaries after improvement.**

| Experiment | IVD Boundary | | | PE Boundary | | |
|---|---|---|---|---|---|---|
| | $d_T = 1$ | $d_T = 2$ | $d_T = 3$ | $d_T = 1$ | $d_T = 2$ | $d_T = 3$ |
| Baseline | 0.68 | 0.87 | 0.90 | 0.79 | 0.91 | 0.94 |
| Proposed 1 | 0.51 | 0.88 | **0.95** | 0.74 | 0.93 | **0.96** |
| Proposed 2 | **0.80** | **0.91** | **0.95** | **0.87** | **0.93** | 0.95 |
| GAC 1 | 0.65 | 0.89 | 0.94 | 0.78 | 0.92 | 0.95 |
| GAC 2 | 0.61 | 0.79 | 0.88 | 0.64 | 0.81 | 0.89 |
| CV 1 | 0.66 | 0.86 | 0.90 | 0.75 | 0.90 | 0.93 |
| CV 2 | 0.67 | 0.86 | 0.90 | 0.70 | 0.87 | 0.92 |

These low baseline BF-Scores, especially when $d_T = 1$, indicate that there is significant room for improvement to be had by applying the contour evolution techniques.

Each contour evolution algorithm is tested in two experimental setups. In the first setup, we compare the contour-evolved manually labeled images vs. unmodified predicted label images produced by a SegNet model trained using the contour-evolved manually labeled images. The second setup compares the contour-evolved manually labeled images vs. contour-evolved predicted label images produced by a SegNet model trained using the contour-evolved manually labeled images. The results of the experiment using these setups are shown in Table 2.

The table shows that when Experiment Setup 1 is employed, all applications of contour evolution methods (Proposed 1, GAC 1, and CV 1) yield worse performance especially when $d_T = 1$. This suggests that improving the boundary only on the manually created label images would have minimal impact on the boundary accuracy of the predicted label images. A marked improvement in the BF-Score occurs when our contour evolution method is applied to both the manually created and predicted label images.

On average, the execution speed of our method when compared to the GAC method at up sampling levels $k = 0$, 1, and 2 are 10.2, 8.6, and 5.2 times faster, respectively. We observed that the main reason why our method is significantly faster is that it carries out a shorter number of iterations before it meets the iteration termination criteria. When compared to the CV method at up sampling levels $k = 0$, 1, and 2, our method is 2.2, 2.1, and 1.5 times faster, respectively. Figs 10, 11 and 12 compare a typical number of pixel changes between successive iterations for each method. As can be seen from these figures, our method solves the bulk of the pixel changes in the first iteration resulting in a much quicker convergence than the GAC method. Although the CV method was shown to have a shorter number of iterations, the method spends more time per iteration than our method which makes it slower than the latter.

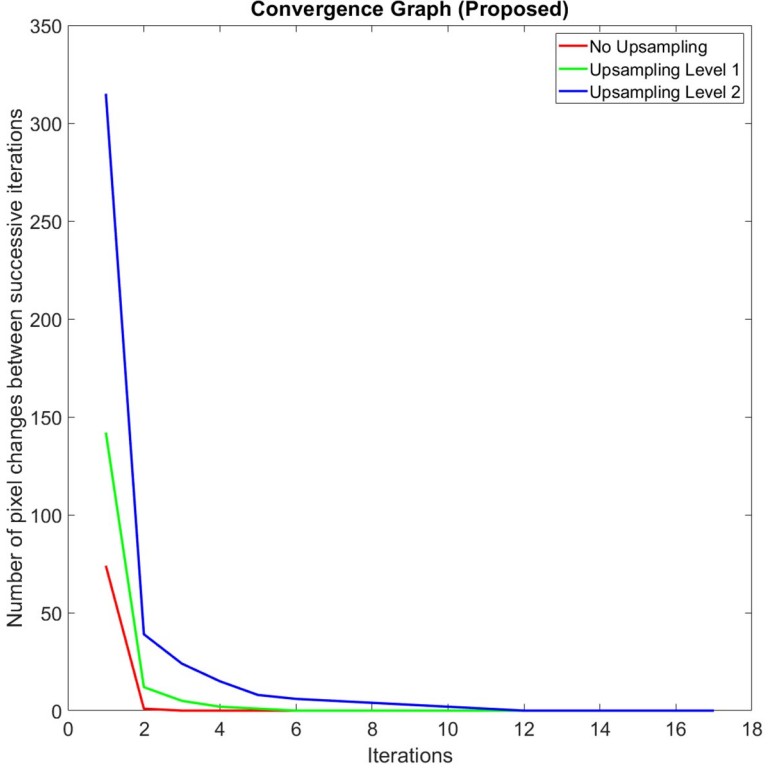

**Fig 10. A typical example of the number of pixel changes between successive iterations of our proposed method.**

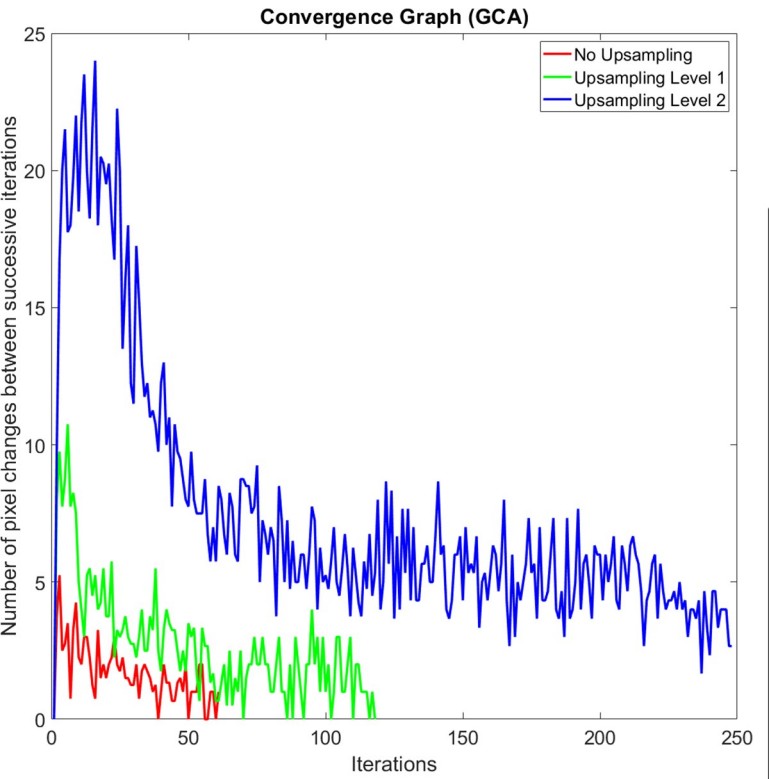

**Fig 11. A typical example of the number of pixel changes between successive iterations of the geodesic active contours method.**

The visual quality of the three contour evolution methods can be seen in Fig 13. We highlighted four cases of boundary changes that exemplify the superiority of our method (Fig 13B) over the others. Case 1 demonstrates a situation where the true boundary is characterized by a significant difference in intensity levels. In this case, our method successfully tightens the boundary line between the darker PE region and the lighter AAP region as do the other two methods. However, a closer inspection of images should show that the CV method (Fig 13D) is less accurate than the other two. Case 2 and 3 demonstrate a situation where the true boundary is characterized by a subtle difference in intensity levels. In both cases, our contour evolution method produces reliably accurate boundaries that are consistent with the manual label. The GAC method (Fig 13C) on the other hand produces inaccurate boundary in case 2 while producing reasonably good boundary in case 3, whereas the CV method produces inaccurate boundary in case 3 while producing reasonably good boundary in case 2. Case 4 shows another artefact of a similar situation. Since we cannot selectively apply GAC or CV method to a specific contour segment, we end up changing the entire boundary. Our contour evolution method, on the other hand, allows this. In our experiment, since we can selectively apply our contour evolution method to only the important IVD and PE boundaries and not to every other region boundaries, those region boundaries such as the one marked 4 can be made unaltered and did not suffer deterioration as shown in Fig 13C and 13D.

## 4.2. Measurement of AP diameter and foraminal widths

In this section, we describe the experiments to assess the suitability of our method in assisting an LSS diagnosis process. The experiments were carried out with the supervision and advice

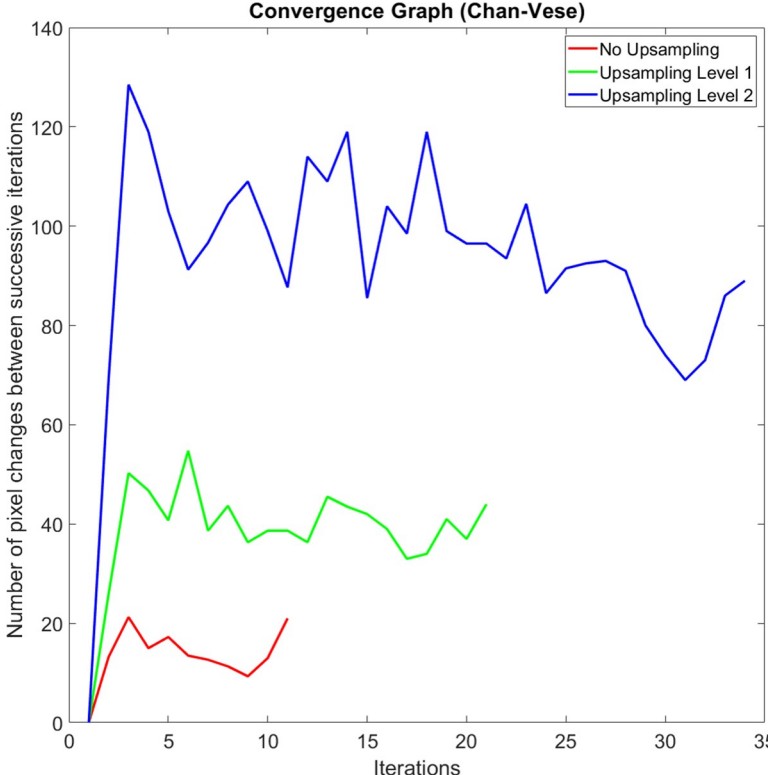

**Fig 12. A typical example of the number of pixel changes between successive iterations of the Chan-Vese method.**

from the expert neuroradiologist members of the team. The objectives of the experiment are as follows:

1. To develop a ground truth of important points/landmarks locations, the AP diameter, the left and right foraminal widths, and the category of the IVD herniation (either normal, minor herniation, and severe herniation) for each patient's L3-L4, L4-L5, and L5-S1 IVDs–a total of 1545 MRI images.

2. To determine the intra-expert variations of the location of the important points in the ground truth. These will be used as an acceptable tolerance of the error between the automatically located points and the ground truth locations.

3. To determine a set of experiment setup that covers a wide combination of experiment parameters.

4. To calculate and compare the performance of the proposed method, the GAC method, and the CV method under each of the above experiment setups.

**4.2.1. Experiment setup.**   To assess the suitability of our methodology, we developed a set of reference dataset containing the location of the six important points, the three measurements, which are the AP diameter, and left and right foraminal widths (in mm), and state of the IVD herniation (normal, minor herniation, and severe herniation). The location of the points is set manually by an expert neuroradiologist in two different settings. Both settings have a common procedure carried out for each MRI image and are described as follows.

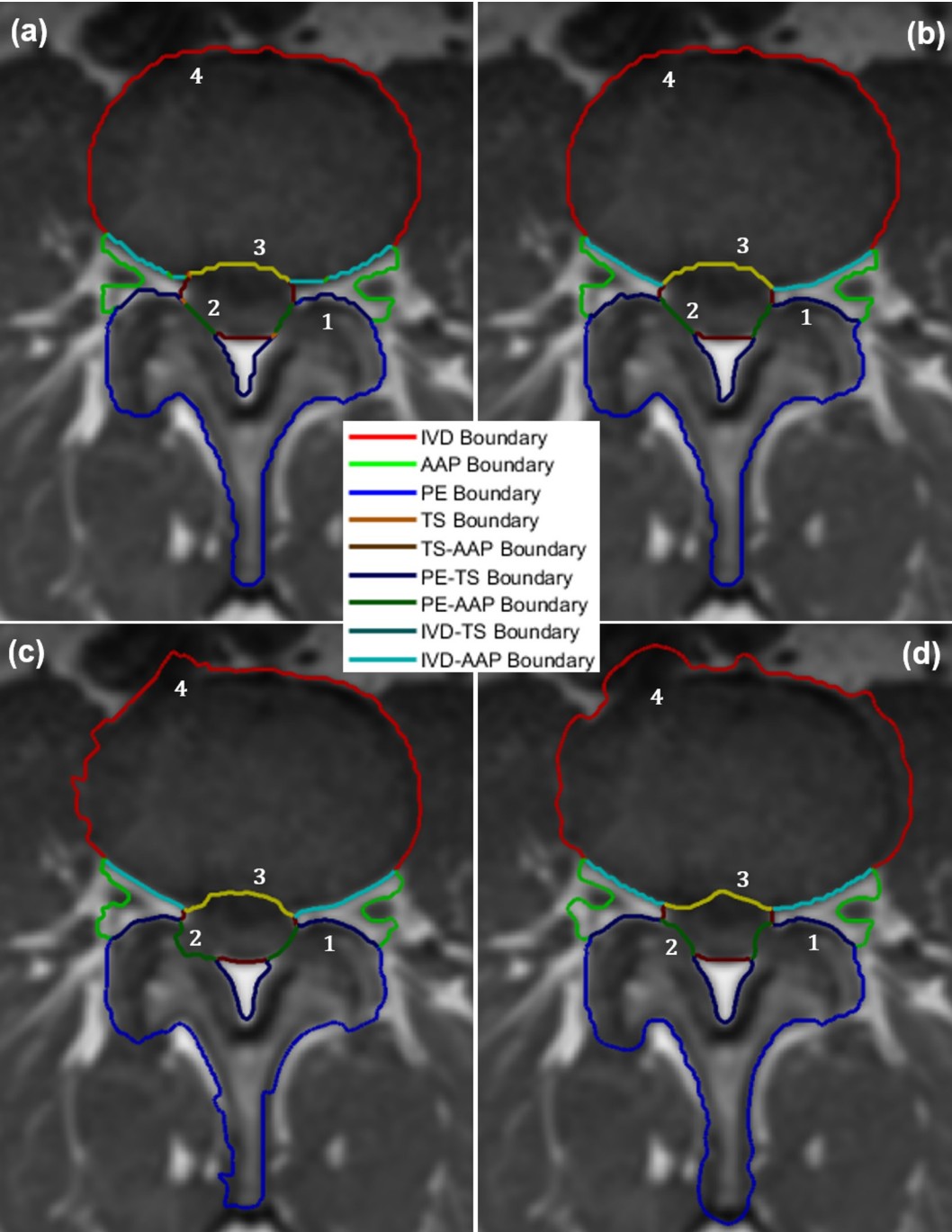

**Fig 13.** An example of the result of the application of a contour evolution method to the a) the original boundary using b) our proposed method, c) Geodesic Active Contours and d) Chan-Vese methods. The numbers mark the location of the four cases that illustrate the superiority of our method to the others.

1. The expert is presented with the MRI image in an axial view then manually locates $T_L$, $T_R$, $B_L$, and $B_R$ points by judging the location of both foraminal widths. The left and right foraminal widths are then automatically calculated.

2. The expert then manually locates $B_M$ and $D_C$ points. Points $p$, $q$, and $q'$ are then automatically determined and the values are used to calculate the AP diameter.

3. The expert can choose to override the calculated AP diameter by manually placing point $q'$.

4. The expert then makes a judgment on the state of the IVD (either normal, minor herniation, or severe herniation).

5. The expert can redo any stage of the procedure to improve his previous judgment.

The whole process is carried out in two different experiment settings. In the first experiment setting (ES1), the expert receives no assistance from our application. The expert was only shown the MRI image to carry out the task. In the second experiment setting (ES2), limited assistance was provided by our application in a form of a segmented image, or region boundary lines, or both, which are superimposed on to the MRI image. This assistance is expected to reduce the difficulty in getting a more precise location of the points once the expert has an idea where they should be. The experiment in each setting was repeated five times, each at different days and times of days resulting in 2×5 sample data points for each important location of the MRI image. The mean location for each point type is computed and the distance of every sample data point to their mean value is calculated. Its average is then used as a measure of the expert's intra-variation in determining the point's location. A small average would indicate higher confidence and vice versa. The intra-expert variations of each of the six types of important points, for both experiment settings, are presented in Table 3.

Following an observation on this result, our analysis can be summarized as follows:

1. There are significant variations in the manual placement of the important points in ES1. This variation is roughly halved in ES2. Since the latter produces less variation, the ES2 result is used as the reference dataset and its variations would be used as the intra-expert variations of the data points in the ground truth.

2. The placement of point $D_C$ has the highest uncertainty because the expert needs to estimate the center of the disc and its location is not on any boundary.

3. The location of the point $B_M$ is relatively easier to decide than the rest, resulting in a lower variation.

As can be seen in Fig 3, our proposed methodology applies a contour evolution technique at two different stages in the workflow. The first application is to the manually labeled images before they are used to train the SegNet model and the second is to the automatically segmented images inferred by the trained SegNet model. To explore the performance of the proposed methodology, we carried out 22 different experiments using a different combination of contour evolution and SegNet applications. In the first six experiments, no SegNet training hence no automatic segmentation takes place. The first experiment locates important boundaries and important points directly from the manually labeled images. The second experiment applies a morphological close operation to the manually labeled images prior to locating important boundaries and important points from them. In the other four experiments, a contour evolution technique is applied to the manually labeled images prior to locating important boundaries and important points from them. In the other sixteen experiments, a SegNet

**Table 3. The intra-expert variations, $\vee$, calculated as the average distance, in mm, between each of the five sample data points and their mean locations.**

| Experiment Setting | $\vee_{TL}$ | $\vee_{BL}$ | $\vee_{TR}$ | $\vee_{BR}$ | $\vee_{BM}$ | $\vee_{DC}$ |
|---|---|---|---|---|---|---|
| ES1 | 1.99 | 2.27 | 1.86 | 2.34 | 1.24 | 2.61 |
| ES2 | 0.89 | 1.10 | 0.83 | 1.17 | 0.76 | 2.20 |

**Table 4. The parameters of each experiment setup.**

| Setup ID | Segmentation method | Training data used | Morphological Close applied? | Contour Evolution Technique |
|---|---|---|---|---|
| M00 | Manually | N/A | No | None |
| M01 | Manually | N/A | Yes | None |
| M10 | Manually | N/A | No | Proposed |
| M11 | Manually | N/A | Yes | Proposed |
| M20 | Manually | N/A | Yes | GAC |
| M30 | Manually | N/A | Yes | CV |
| A10 | Automatically | M00 | No | None |
| A11 | Automatically | M00 | Yes | None |
| A12 | Automatically | M00 | No | Proposed |
| A13 | Automatically | M00 | Yes | Proposed |
| A20 | Automatically | M11 | No | None |
| A21 | Automatically | M11 | Yes | None |
| A22 | Automatically | M11 | No | Proposed |
| A23 | Automatically | M11 | Yes | Proposed |
| A30 | Automatically | M20 | No | None |
| A31 | Automatically | M20 | Yes | None |
| A32 | Automatically | M20 | No | GAC |
| A33 | Automatically | M20 | Yes | GAC |
| A40 | Automatically | M30 | No | None |
| A41 | Automatically | M30 | Yes | None |
| A42 | Automatically | M30 | No | CV |
| A43 | Automatically | M30 | Yes | CV |

model is trained and is used to automatically segment the MRI images. These sixteen experiments are grouped into four sets of four experiments, with each group uses one of four different SegNet models. Each SegNet model is trained using either unmodified manual label images, modified manual label images using the proposed method, modified manual label images using the GAC method, or modified manual label images using the CV method. Each of the four experiments applies the processes to the automatically segmented images in a different setup, which are a combination of a) whether or not a morphological close operation is applied and b) whether or not the respective contour evolution technique is applied. The overall setup of the 22 experiments is summarized in Table 4. Each setup is assigned a unique ID for ease of reference in subsequent discussions.

**4.2.2. Experiment results and analysis.** The result of the 22 experiments is tabulated in Table 5. The table shows the mean error (in mm), averaged over 1545 images, between the predicted and the reference locations of points $T_L$, $T_R$, $B_L$, $B_R$, $B_M$, and $D_C$. It also shows the mean error (in mm) between the predicted left and right foraminal widths and AP diameter and their corresponding values calculated using the reference important points location. The last column of the table shows the agreement score with an expert opinion on the severity of the IVD herniation, which is the percentage of correctly predicted IVD herniation state over 1545 images.

Predictions that have lower error than the intra-expert variations are shaded in yellow whereas the best results are marked in bold.

Our analysis of this result can be summarized as follows:

1. Under an identical setting, our proposed contour evolution method consistently produces lower errors than when no contour evolution method is applied.

**Table 5. Mean error in mm, ε, between each predicted important point location and its reference location (column 2–7), between the predicted left and right foraminal widths and AP diameter and their corresponding values (column 8–10) and the IVD herniation diagnosis accuracy in % (last column).**

| ID | $\varepsilon_{TL}$ | $\varepsilon_{BL}$ | $\varepsilon_{TR}$ | $\varepsilon_{BR}$ | $\varepsilon_{BM}$ | $\varepsilon_{DC}$ | $\varepsilon_{dL}$ | $\varepsilon_{dR}$ | $\varepsilon_{dAP}$ | $a_D$ |
|---|---|---|---|---|---|---|---|---|---|---|
| M00 | 1.95 | 1.70 | 1.84 | 1.69 | 0.95 | 1.48 | 0.8 | 0.7 | 1.13 | 95.1 |
| M01 | 1.74 | 1.52 | 1.52 | 1.43 | 0.77 | 1.45 | 0.64 | 0.51 | 0.92 | 95.5 |
| M10 | 1.42 | 1.26 | 1.37 | 1.23 | 0.64 | 1.49 | 0.50 | 0.49 | 0.87 | 95.5 |
| M11 | 1.19 | 1.06 | 1.13 | 1.00 | 0.63 | 1.50 | 0.41 | 0.43 | 0.84 | 95.8 |
| M20 | 2.22 | 1.95 | 1.82 | 1.73 | 2.37 | 1.83 | 0.78 | 0.71 | 3.07 | 91.7 |
| M30 | 1.95 | 1.70 | 1.84 | 1.69 | 0.95 | 1.48 | 0.80 | 0.70 | 1.13 | 95.1 |
| A10 | 1.75 | 1.53 | 1.54 | 1.44 | 1.03 | 1.41 | 0.85 | 0.69 | 1.42 | 95.4 |
| A11 | 1.78 | 1.57 | 1.51 | 1.38 | 0.90 | 1.42 | 0.85 | 0.61 | 1.15 | 95.5 |
| A12 | 0.99 | 0.88 | 1.00 | 0.89 | 0.72 | 1.39 | 0.37 | 0.37 | 1.10 | 95.6 |
| A13 | 1.01 | 0.90 | 1.01 | 0.89 | 0.73 | 1.4 | 0.37 | 0.37 | 1.10 | 95.7 |
| A20 | 1.59 | 1.40 | 1.37 | 1.35 | 0.97 | 1.36 | 0.79 | 0.65 | 1.36 | 95.5 |
| A21 | 1.59 | 1.39 | 1.36 | 1.28 | 0.79 | 1.37 | 0.79 | 0.57 | 1.02 | 95.9 |
| A22 | **0.41** | **0.41** | **0.37** | **0.37** | **0.53** | **1.36** | **0.29** | **0.28** | **0.90** | 96.7 |
| A23 | **0.41** | 0.42 | **0.37** | 0.38 | 0.54 | 1.36 | 0.29 | 0.28 | 0.91 | **96.6** |
| A30 | 2.21 | 1.93 | 1.97 | 1.85 | 2.05 | 2.16 | 1.05 | 1.00 | 2.66 | 93.6 |
| A31 | 2.21 | 1.93 | 1.97 | 1.85 | 2.07 | 2.18 | 1.05 | 1.01 | 2.65 | 93.8 |
| A32 | 3.50 | 3.13 | 3.16 | 2.94 | 4.13 | 3.96 | 1.44 | 1.37 | 5.26 | 87.0 |
| A33 | 3.44 | 3.09 | 3.16 | 2.95 | 4.12 | 3.97 | 1.44 | 1.39 | 5.29 | 86.9 |
| A40 | 1.77 | 1.54 | 1.6 | 1.51 | 0.89 | 1.42 | 0.85 | 0.73 | 1.15 | 96.1 |
| A41 | 1.77 | 1.54 | 1.59 | 1.50 | 0.89 | 1.43 | 0.85 | 0.73 | 1.15 | 96.2 |
| A42 | 1.81 | 1.61 | 1.59 | 1.54 | 1.24 | 1.5 | 0.77 | 0.67 | 1.23 | 95.7 |
| A43 | 1.80 | 1.59 | 1.57 | 1.51 | 1.22 | 1.48 | 0.77 | 0.67 | 1.22 | 95.8 |

2. Our proposed contour evolution method performs significantly better than GAC and CV methods when they are applied under an identical setting.

3. The application of morphological close operation produces marginal but inconsistent improvement.

4. Most setups yield less variation of $D_C$ point location than the intra-expert variation with the exception of A32 and A33.

5. The best performing setup to measure the left and right foraminal distances uses our proposed contour evolution method on automatically segmented label images using a SegNet model trained using manually segmented label images that have been improved using the same method. Using this setup, the average error of the calculated right and left foraminal distances relative to their expert-measured distances are 0.28 mm ($p = 0.92$) and 0.29 mm ($p = 0.97$), respectively. The value in bracket is the $p$-value of a one-sample t-test on the result.

6. Using the same setup, the average error of the calculated AP diameter relative to their expert-measured diameter is 0.90 mm ($p = 0.92$), which is slightly less accurate than the best setup, i.e., using our proposed contour evolution method on manually segmented label images (0.84 mm average error).

7. Our method produces the best improvement by improving the IVD herniation diagnosis accuracy performance by 1.6 percentage points (96.7% accuracy compared to 95.1% accuracy using only the manually segmented label images).

Fig 14 shows the visual interface of the developed computer-aided diagnostic containing the results of the boundary delineation, foraminal widths measurements, and central IVD herniation classification. Fig 14A shows a healthy L3-L4 disk whereas Fig 14B shows a case where

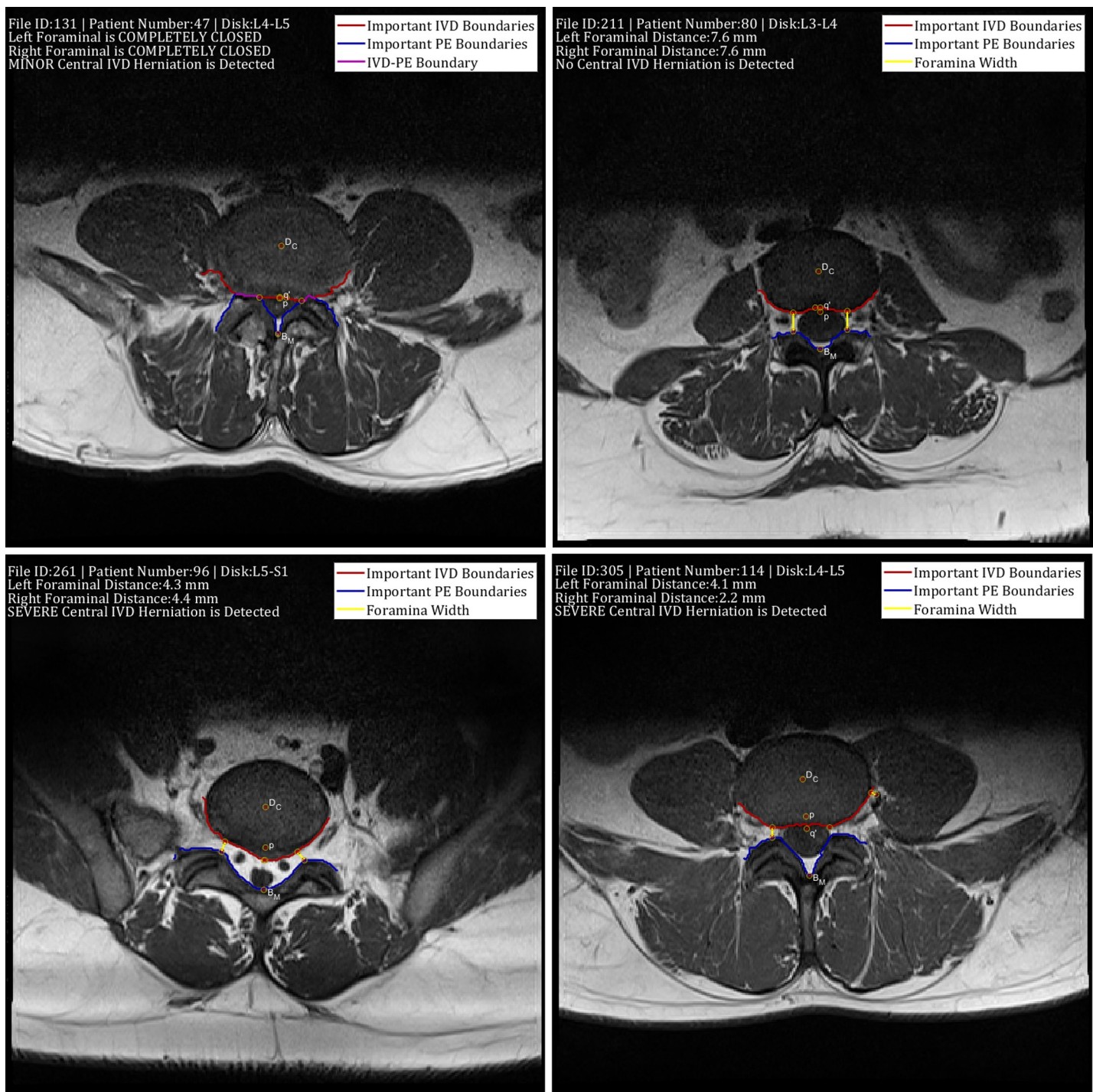

**Fig 14. The visual interface of the developed computer-aided diagnostic system showing the results boundary delineation results, foraminal widths measurements, and central IVD herniation classification.** The figure shows a) a healthy L3-L4 disk, b) and c) IVDs with different types of abnormalities and d) a case where the right foraminal width measurement and diagnostic results are erroneous.

the IVD touches the PE causing a complete closing of the lateral foramina gap. to d) IVDs with different types of abnormalities. Fig 14C shows a case where a severe central IVD herniation is detected. Fig 14D, on the other hand, shows a situation where misclassification of several 'Other' region pixels as belonging to the PE region causing a problem with the right foraminal width measurement.

## 5. Conclusion

We have presented in this paper a novel method to help neuroradiologists diagnose lumbar spinal stenosis by automatic measurement of the AP diameter and foraminal widths that are calculated using nine important points on an axial MRI image. The points were located after the automatic segmentation of the image using SegNet and improving important region boundaries using our contour evolution technique.

The performance of the proposed algorithm was evaluated through a set of experiments on our publicly available Lumbar Spine MRI dataset containing MRI studies of 515 patients with symptomatic back pains. Our experiments show that our method can locate all the important points that are on region boundaries accurately. This allows accurate calculation of the AP diameter, which in turn, produces high agreement with the expert's opinion when determining the severity of the IVD herniations.

## Supporting information

**S1 File. A Word document containing Algorithm 1 and Algorithm 2.**
(DOCX)

**S2 File. A Word document containing URLs and DOIs of the dataset and corresponding MATLAB code to reproduce the results.**
(DOCX)

## Author Contributions

**Conceptualization:** Ala Al-Kafri, Sud Sudirman.

**Data curation:** Reyhan Eddy Yunus, Mohammed Al-Jumaily.

**Funding acquisition:** Friska Natalia.

**Investigation:** Friska Natalia.

**Methodology:** Friska Natalia, Hira Meidia, Sud Sudirman.

**Project administration:** Friska Natalia.

**Software:** Nunik Afriliana, Julio Christian Young.

**Supervision:** Sud Sudirman.

**Validation:** Reyhan Eddy Yunus, Mohammed Al-Jumaily.

**Writing – original draft:** Ala Al-Kafri.

**Writing – review & editing:** Friska Natalia, Sud Sudirman.

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
