## [Decision Letter · Decision Letter 0]

3 Jun 2020

PONE-D-20-11502

Automated measurement of anteroposterior diameter and foramina widths in MRI images for lumbar spinal stenosis diagnosis

PLOS ONE

Dear Dr. Sudirman,

Thank you for submitting your manuscript to PLOS ONE. After careful consideration, we feel that it has merit but does not fully meet PLOS ONE’s publication criteria as it currently stands. Therefore, we invite you to submit a revised version of the manuscript that addresses the points raised during the review process.  Both reviewers felt there were aspects of the methodology that required clarification.  Reviewer 1 additionally suggested that restructuring parts of the paper would improve its readability. 

We look forward to receiving your revised manuscript.

Kind regards,

Dzung Pham

Academic Editor

PLOS ONE

Journal Requirements:

2. Please clarify whether any data was collected specifically for this study.

Does the ethics approval apply specifically for this study, or to the original data collection?

Reviewers' comments:

Reviewer's Responses to Questions

**Comments to the Author**

1. Is the manuscript technically sound, and do the data support the conclusions?

Reviewer #1: Partly

Reviewer #2: Yes

2. Has the statistical analysis been performed appropriately and rigorously? 

Reviewer #1: No

Reviewer #2: Yes

3. Have the authors made all data underlying the findings in their manuscript fully available?

Reviewer #1: Yes

Reviewer #2: Yes

4. Is the manuscript presented in an intelligible fashion and written in standard English?

Reviewer #1: No

Reviewer #2: Yes

5. Review Comments to the Author

Reviewer #1: The paper presented a workflow to segment the LSS regions of interest and locate the landmarks to evaluating the pathologies. A contour evolution algorithm is proposed to improve the segmentation.

The Reviewer's major reservation is the paper structure and presentation. Significantly major work is required to improve the paper structure and language. For example, section 3. Mehod contains a mix of literature review, methodology and results. For example Table 1, 2, 4 are results, but placed in Method section. If they are new results of this paper, please move to Result section. If they are results from previous work, please replace with references.

The method contribution of this paper is not clear. Do the Authors only contribute to a contour evolution techniuque? If this is the case then the title and introduction need major change to reflect the content of the paper. Please provide reference to the existing methods at their first appearance, for example, Line 170, SegNet model.

L108 "He or she then mentally locates 109 several regions of interest (ROIs)", mentally -> manually?

L167, "Image registration is applied to align each T1 and T2 pair, and a composite

168 image is produced from them. ", no detail on image registration method.

L169, 'split into a training and a test set with an 80:20 ratio', 80% and 20%

L170, 'develop a SegNet model': please provide details or direct reference.

L195, 'Interested readers are encouraged to refer to the paper to get a more detailed description of the SegNet architecture that we used.' Method in a paper should be self-explaining and should provide sufficient information to the readers without referring to other work. Please remove the sentence.

L198: 'The SegNet model is trained using 80% of the dataset with no cross-validation' This seems to be duplication of L169.

L218: 'with an 80:20 training-to-testing ratio', again this is duplication of L198 etc.

Table 1; What is the purpose of segmentatin 'unregistered' area? If not of relevance, plese remove.

L248 - 252, wordy and difficult to follow. Please rephase.

L267 - 281, 'We have previously shown (Natalia et al., 2019b; Young et al., 2019) the two

drawbacks in existing active contour models ...' This is literature review. Please move to backgound or related work section.

L624, Conclusion should be breif and convey the important and conclusive messages. Plesae shortening.

Reviewer #2: Really great work. Very thorough and in depth. I have several questions and suggestions:

Line 169: Is the split done on per patient basis?

Lines 198-200: How is the composite image constructed; is it a 2 channel image, one channel for T1 and one channel for T2?

Line 215: VGG16 network trained on ImageNet takes in a 3 channel image as input; how is the network modified to take in the composite image? Do you have to retrain the first layer of the VGG16 network?

Line 224: It seems that the `unregistered` class is a bit unnecessary; is there any reason to try and predict it as a separate class instead of combining it with `other`?

Table 2 is unnecessary since the first row of Table 4 shows the same thing.

Table 7: How is herniation defined? Is it a binary classification task?

6. PLOS authors have the option to publish the peer review history of their article (what does this mean?). If published, this will include your full peer review and any attached files.

Reviewer #1: No

Reviewer #2: No

---

## [Author Response · Author response to Decision Letter 0]

1 Jul 2020

First of all, thank you to both reviewers for kindly reviewing our paper. Your comments are very valuable for improving the quality of the paper. We have made the necessary changes to the paper based on your suggestions and questions. Please find below our response to specific areas of the paper and the outline of the changes that we made.

Reviewer #1: The paper presented a workflow to segment the LSS regions of interest and locate the landmarks to evaluating the pathologies. A contour evolution algorithm is proposed to improve the segmentation.

1. The Reviewer's major reservation is the paper structure and presentation. Significantly major work is required to improve the paper structure and language. For example, section 3. Method contains a mix of literature review, methodology and results. For example, Table 1, 2, 4 are results, but placed in Method section. If they are new results of this paper, please move to Result section. If they are results from previous work, please replace with references.

Thank you for your suggestion. We have made significant changes to our paper. These can be summarised as follows:

• Moving background information and related approaches on image segmentation from various parts of the paper to the end of Section 2 (line 142-165).

• We added additional information (~500 words: line 184-224) to describe the proposed methodology to make it clearer to the readers. This includes more explicit description on the role of our previous findings (SegNet segmentation and contour evolution) in the overall methodology and more verbose description on the image registration process.

• Removing any intermediate results from Section 3. A majority of the information (including Table 4) were moved to Section 4, some were removed due to redundancy, and some (Tables 1 & 2) were replaced with references to where it first appeared.

• Section 4 now consists of two sub-sections, one to discuss the Segmentation and Contour Evolution experiment and results (line 449-553) and the other to discuss the AP diameter and foraminal width measurements experiment and results (line 554-690).

• Additional information provided on the meaning of the quoted p-value (line 672-673)

• Shortening of the conclusion to make it more concise (line 691-706).

We have also improved the English quality by having the latest version professionally proofread.

2. The method contribution of this paper is not clear. Do the Authors only contribute to a contour evolution technique? If this is the case, then the title and introduction need major change to reflect the content of the paper. Please provide reference to the existing methods at their first appearance, for example, Line 170, SegNet model.

The paper novel contribution is not only on the contour evolution technique but the entire workflow as illustrated in Figure 3. Granted that several parts of the workflow have been introduced in our previous publications, but they were proposed specifically to solve a specific problem in the more general context of lumbar spinal stenosis diagnosis. For example, in (Al-Kafri et al., 2019) we addressed the SegNet model development for semantic image segmentation. The training data used to develop this model is unmodified manually segmented images. In (Natalia et al., 2020) we developed a technique to improve the accuracy of delineated boundaries of the segmented images. Our experiment on using this technique on the manually labeled images shown significant visual improvement on the accuracy of the segmentation along important region borders. In this paper, we combine these techniques by first using the improved manually labeled images (training set) to retrain the SegNet model. We then use the retrained SegNet model to automatically segment lumbar spine MRI images (test set). We then reapply the contour evolution technique on the resulting segmented images to further improve the results. Using the technique that we proposed in (Natalia et al., 2019), we use the resulting segmented images to locate important points that are subsequently used to measure the anteroposterior diameter and foramina widths and detect an occurrence of IVD herniation.

We hope this makes the rationale of the methodology in this paper and its relationship with our previous works. On the paper itself, we have made several changes (line 184-224) to make the paper contribution much clearer. Please accept our apology if this was not clear in the previous version.

3. L108 "He or she then mentally locates 109 several regions of interest (ROIs)", mentally -> manually?

Apology for the confusion. When we wrote this, we meant that the neuroradiologist locates the regions of interest in their mind only without making annotation or marker on the regions’ boundary (just like how mental maths is done). We thought that the correct word was mentally, but we understand now that the word is not accurate. Instead, we will use your suggestion to replace it with the word manually. 

4. L167, "Image registration is applied to align each T1 and T2 pair, and a composite image is produced from them". no detail on image registration method.

Additional information (line 184-199) is added to describe the image registration process. This reads:

“We then apply an image registration to each T1- and T2-weighted image pair to ensure that every pixel at the same location in both images corresponds to the same point in an organ or tissue. This is performed by calculating the difference between a fixed T1-weighted image and a set of transformed T2-weighted image produced over a search-space of affine transforms. The minimum and maximum limit of the radius of the search-space is set to 1.5e-6 and 13e-3 respectively, and with a growth factor of 1.05 over 300 iterations. To counter the effect of high-frequency noise and low-frequency inhomogeneity field on both modalities, a parametric bias field estimation is applied before being corrected using PABIC method (Styner et al., 2000). A search optimization algorithm called (1+1)-Evolutionary Strategy is employed by locally adjusting the search direction and step size while at the same time provides a mechanism to step out of non-optimal local minima. After the registration process completes, a composite 3-channel image is created using the fixed T1-weighted and transformed T2-weighted images for the first and second channels. The third channel is set using the Manhattan distance of the two images.”

5. L169, 'split into a training and a test set with an 80:20 ratio', 80% and 20%

This sentence has been reworded into, “The resulting 1545 composite images are then randomly split into a training and a test set, containing 80% and 20% of the dataset, respectively.” (line 200-201)

6. L170, 'develop a SegNet model': please provide details or direct reference.

Our response to this relates very closely to your next question, so please refer to our response to that below. Additionally, we have also included the reference (Badrinarayanan, Kendall and Cipolla, 2017) to the original paper in which the SegNet model was first proposed (line 202).

7. L195, 'Interested readers are encouraged to refer to the paper to get a more detailed description of the SegNet architecture that we used.' Method in a paper should be self-explaining and should provide sufficient information to the readers without referring to other work. Please remove the sentence.

This sentence has been removed and in its place several sentences were added to describe the relationship of our previous work to the method described in this paper. They are: 

“We have previously shown the applicability of the SegNet architecture in segmenting axial MRI images using unmodified label images (Al-Kafri et al., 2019). We will use the previously reported results as a baseline to measure the improvement in the segmentation accuracy along the important boundaries using the proposed method” (line 218-221)

“We adopted the transfer learning approach (Tan et al., 2018) when training the SegNet model instead of developing the model from scratch since we have shown previously that the former approach produces significantly better segmentation than the latter approach (Al-Kafri et al., 2019). We did this by using a SegNet model with pre-trained VGG16 coefficients on the ImageNet database (Deng et al., 2009). The classification layer was replaced with a new randomly initialized fully connected neural network before the whole network is retrained using the training dataset.

During the training process, the SegNet model’s weights and biases are updated using the popular Stochastic Gradient Descent with Momentum algorithm (Ruder, 2016) with a learning rate of 0.001. The size of the training batch is 40 and the maximum number of epochs is 100. The input images need to be resized to 360�480 to match the input size requirement of the VGG16 network. Furthermore, to balance the importance of all classes due to class population imbalance, we apply class weighting (Zhou and Liu, 2010). To make sure that small-sized classes, such as the TS and AAP, are not underrepresented in our training data we set the class weighting to be inversely proportional to the class population. The segmented images produced by the trained model are then resized back to 320�320 before being used in the rest of the workflow.” (line 451-467)

“This is shown in the first row of Table 2. The result shows that only 68% of the important IVD boundary points and 79% of the important PE boundary points of the segmented images are correctly located within one pixel of their actual locations. These low baseline BF-Scores, especially when dT = 1, indicate that there is significant room for improvement to be had by applying the contour evolution techniques.” (line 496-501)

8. L198: 'The SegNet model is trained using 80% of the dataset with no cross-validation' This seems to be duplication of L169.

This sentence has been removed to avoid duplication.

9. L218: 'with an 80:20 training-to-testing ratio', again this is a duplication of L198, etc.

This part of a sentence has also been removed.

10. Table 1; What is the purpose of segmentation 'unregistered' area? If not of relevance, please remove.

This is admittedly a very difficult issue to explain. The reason for the inclusion of the ‘Unregistered’ region to the classification model is because we did the same in our previous paper (Al-Kafri et al., 2019) when we described the the feasibility of SegNet to segment and delineate lumbar spine MRI images. The reason at that time is because the property of the 3-channel (T1, T2, and Manhattan distance) pixels of the ‘Unregistered’ region differs to those of the ‘Other’ region. It is true that there is not much use of differentiating the two for the purpose of delineating the important region boundaries, however, since we are using our previous results for comparison, we think that to be consistent we should keep the SegNet structure the same. 

I hope this explanation is sufficient to keep the ‘Unregistered’ region as part of the image segmentation process.

11. L248 - 252, wordy and difficult to follow. Please rephase.

This paragraph explains Table 2 which is part of the experiment result that was included in the Material and Method section. The information that was contained in the table is the same as the first row of Table 4 in the result section, Hence, due to the restructuring and as the suggestion of the other reviewer to remove information redundancies, Table 2 has been removed. Since this sentence was meant to provide a rationale on the need to apply contour evolution based on the previous experiment result, the sentence was moved to Section 4 and rephrased as follows:

“The result shows that only 68% of the important IVD boundary points and 79% of the important PE boundary points of the segmented images are correctly located within one pixel of their actual locations. These low baseline BF-Scores, especially when dT = 1, indicate that there is significant room for improvement to be had by applying the contour evolution techniques” (line 497-501).

12. L267 - 281, 'We have previously shown (Natalia et al., 2019b; Young et al., 2019) the two drawbacks in existing active contour models ...' This is a literature review. Please move to background or related work section.

This part has been moved to Section 2 and rephrased. It now reads:

“We have, however, previously shown (Young et al., 2019; Natalia et al., 2020) two drawbacks in existing active contour models. Firstly, existing approaches cannot apply contour evolution to only specific parts of the contour. Having the ability to decide which part of the contour to evolve and which feature to use, would allow for a more adaptive contour evolution. Secondly, the inclusion of all control parameters when calculating the energy function minimization makes it harder to find the right combination that yields the best results. The effect of adjusting one control parameter may counter the effect of adjusting the others which results in finding the right combination of parameter values involves trial-and-error and can be very tricky and case dependent” (line 153-161).

13. L624, Conclusion should be brief and convey the important and conclusive messages. Please shortening.

The conclusion is shortened and made more concise. It now reads:

“We have presented in this paper a novel method to help neuroradiologists diagnose lumbar spinal stenosis by automatic measurement of the AP diameter and foramina widths that are calculated using nine important points on an axial MRI image. The points were located after the automatic segmentation of the image using SegNet and improving important region boundaries using our contour evolution technique.

The performance of the proposed algorithm was evaluated through a set of experiments on our publicly available Lumbar Spine MRI dataset containing MRI studies of 515 patients with symptomatic back pains. Our experiments show that our method can locate important points that are on region boundaries between 0.37 – 1.36 mm of the expert-located reference points. The average error of the calculated right and left foraminal distances relative to their expert-measured distances are 0.28 mm (p = 0.92) and 0.29 mm (p = 0.97), respectively. The average error of the calculated AP diameter relative to their expert-measured diameter is 0.90 mm (p = 0.92). The method also achieves 96.7% agreement with an expert opinion on determining the severity of the IVD herniation” (line 692-706).

 

Reviewer #2: Really great work. Very thorough and in depth. I have several questions and suggestions:

1. Line 169: Is the split done on a per-patient basis?

First of all, we would like to thank you for your review. We really appreciate your comments and found them really encouraging.

As for this specific question, the dataset split was done not on per-patient basis but as a whole dataset. This study uses 1545 images from 515 patients � 3 IVDs. The split is done randomly on these 1545 images. 

2. Lines 198-200: How is the composite image constructed; is it a 2 channel image, one channel for T1 and one channel for T2?

Each composite image consists of three channel, the T1-, the aligned T2- and the Manhattan difference of the two. Apology for not making this clearer previously. We added a sentence on the paper to describe the composite image after describing the image registration process. The sentence reads:

“After the registration process completes, a composite 3-channel image is created using the fixed T1-weighted and transformed T2-weighted images for the first and second channels. The third channel is set to the Manhattan distance of the two images.” (line 196-199)

3. Line 215: VGG16 network trained on ImageNet takes in a 3 channel image as input; how is the network modified to take in the composite image? Do you have to retrain the first layer of the VGG16 network?

Our input image also consists of three channels (T1-, aligned T2-, and Manhattan difference). We do however resized the image dimension to 480�360 before training and inferencing, and resized it back to 320�320 before we proceed with the rest of the workflow. We added a sentence when we describe the SegNet training setup to 

“The input images need to be resized to 360�480 to match the input size requirement of the VGG16 network.” (line 461-462)

“The segmented images produced by the trained model are then resized back to 320�320 before being used in the rest of the workflow.” (line 466-467)

4. Line 224: It seems that the `unregistered` class is a bit unnecessary; is there any reason to try and predict it as a separate class instead of combining it with `other`?

This is admittedly a very difficult issue to explain – and incidentally was also brought up by the other reviewer. The reason for the inclusion of the ‘Unregistered’ region to the classification model is because we did the same in our previous paper (Al-Kafri et al., 2019) when we described the the feasibility of SegNet to segment and delineate lumbar spine MRI images. At that time, it was suggested by one of the reviewers of the paper that the property of the 3-channel (T1, T2, and Manhattan distance) pixels of the ‘Unregistered’ region differs to those of the ‘Other’ region – hence would benefit from the separation. It is true that there is not much use of differentiating the two for the purpose of delineating the important region boundaries, however, since we are using our previous results for comparison, we think that to be consistent we should keep the SegNet structure the same. 

I hope this explanation is sufficient to keep the ‘Unregistered’ region as part of the image segmentation process.

5. Table 2 is unnecessary since the first row of Table 4 shows the same thing.

Table 2 has been removed from the paper. The sentence that described the table has been replaced with:

“This is shown in the first row of Table 2. The result shows that only 68% of the important IVD boundary points and 79% of the important PE boundary points of the segmented images are correctly located within one pixel of their actual locations. These low baseline BF-Scores, especially when dT = 1, indicate that there is significant room for improvement to be had by applying the contour evolution techniques.” (line 496-501)

6. Table 7: How is herniation defined? Is it a binary classification task?

The task categorises each case into one of the three namely Healthy, Minor Herniation and Severe Herniation depending on the relative position of the important IVD boundary line with respect to the horizontal line connecting the top lateral points. We have added description on this on the paper:

“A healthy kidney-shaped IVD would result in r � 1.1, which means point q� is sufficiently above the TL-TR line segment. A herniated IVD, on the other hand, would result in smaller values of r. We further classify this case into two sub-categories, namely minor and severe herniations. A severe IVD herniation is detected when dq� is much shorter than dp and minor herniation is when dq� is slightly shorter or about the same length as dp. We use a threshold value of 0.8 to differentiate the two instances, i.e., severe herniation when r � 0.8 and minor herniation when 0.8 < r < 1.1.” (line 434-441)

The idea was inferred originally from (Yates, Giangregorio and McGill, 2010) and translated to a mathematical model for computer implementation in (Natalia et al., 2019). To help show the readers how the system presents the measurement and diagnosis results, we have added one additional figure (Figure 14). The description as included in the paper is as follows:

“Figure 14 shows the visual interface of the developed computer-aided diagnostic containing the results of the boundary delineation, foraminal widths measurements, and central IVD herniation classification. Figure 14a shows a healthy L3-L4 disk whereas Figure 14b shows a case where the IVD touches the PE causing a complete closing of the lateral foramina gap. to d) IVDs with different types of abnormalities. Figure 14c shows a case where a severe central IVD herniation is detected. Figure 14d, on the other hand, shows a situation where misclassification of several ‘Other’ region pixels as belonging to the PE region causing a problem with the right foraminal width measurement.” (line 681-689)

References

Al-Kafri, A. S., Sudirman, S., Hussain, A., Al-Jumeily, D., Natalia, F., Meidia, H., Afriliana, N., Al-Rashdan, W., Bashtawi, M. and Al-Jumaily, M. (2019) ‘Boundary Delineation of MRI Images for Lumbar Spinal Stenosis Detection Through Semantic Segmentation Using Deep Neural Networks’, IEEE Access, 7, pp. 43487–43501. doi: 10.1109/ACCESS.2019.2908002.

Badrinarayanan, V., Kendall, A. and Cipolla, R. (2017) ‘SegNet: A Deep Convolutional Encoder-Decoder Architecture for Image Segmentation’, IEEE Transactions on Pattern Analysis and Machine Intelligence, 39(12), pp. 2481–2495. doi: 10.1109/TPAMI.2016.2644615.

Deng, J., Dong, W., Socher, R., Li, L.-J., Li, K. and Fei-Fei, L. (2009) ‘Imagenet: A large-scale hierarchical image database’, in Computer Vision and Pattern Recognition, 2009. CVPR 2009. IEEE Conference on, pp. 248–255.

Natalia, F., Meidia, H., Afriliana, N., Al-Kafri, A. and Sudirman, S. (2019) ‘Methodology to Determine Important-Points Location for Automated Lumbar Spine Stenosis Diagnosis Procedure’, in International Conference on Intelligent Medicine and Health (ICIMH). Ningbo, China, pp. 53–57. doi: 10.1145/3348416.3348426.

Natalia, F., Meidia, H., Afriliana, N., Young, J. C. and Sudirman, S. (2020) ‘Contour evolution method for precise boundary delineation of medical images’, Telkomnika (Telecommunication Computing Electronics and Control), 18(3), pp. 1621–1632. doi: 10.12928/TELKOMNIKA.v18i3.14746.

Ruder, S. (2016) ‘An overview of gradient descent optimization algorithms’, CoRR, abs/1609.0. Available at: http://arxiv.org/abs/1609.04747.

Styner, M., Brechbuhler, C., Szckely, G. and Gerig, G. (2000) ‘Parametric estimate of intensity inhomogeneities applied to MRI.’, IEEE Trans. Med. Imaging, 19(3), pp. 153–165. doi: 10.1109/42.845174.

Tan, C., Sun, F., Kong, T., Zhang, W., Yang, C. and Liu, C. (2018) ‘A survey on deep transfer learning’, in International conference on artificial neural networks, pp. 270–279.

Yates, J. P., Giangregorio, L. and McGill, S. M. (2010) ‘The influence of intervertebral disc shape on the pathway of posterior/posterolateral partial herniation’, Spine. LWW, 35(7), pp. 734–739.

Young, J. C., Afriliana, N., Natalia, F., Meidia, H. and Sudirman, S. (2019) ‘A Study on the Suitability of Applying Active Contour Evolution Models in Segmenting and Delineating Boundaries in Medical Images’, in 5th International Conference on New Media Studies (CONMEDIA). Denpasar, Bali: IEEE, pp. 232–237. doi: 10.1109/CONMEDIA46929.2019.8981855.

Zhou, Z.-H. and Liu, X.-Y. (2010) ‘On multi-class cost-sensitive learning’, Computational Intelligence. Wiley Online Library, 26(3), pp. 232–257.

---

## [Decision Letter · Decision Letter 1]

8 Oct 2020

PONE-D-20-11502R1

Automated measurement of anteroposterior diameter and foraminal widths in MRI images for lumbar spinal stenosis diagnosis

PLOS ONE

Dear Dr. Sudirman,

Thank you for submitting your manuscript to PLOS ONE. After careful consideration, we feel that it has merit but does not fully meet PLOS ONE’s publication criteria as it currently stands. Therefore, we invite you to submit a revised version of the manuscript that addresses the points raised during the review process.  In particular, as raised by Reviewer 1, please remove specific results from the Conclusion that were already provided in the abstract and elsewhere in the manuscript.

We look forward to receiving your revised manuscript.

Kind regards,

Dzung Pham

Academic Editor

PLOS ONE

Reviewers' comments:

Reviewer's Responses to Questions

**Comments to the Author**

1. If the authors have adequately addressed your comments raised in a previous round of review and you feel that this manuscript is now acceptable for publication, you may indicate that here to bypass the “Comments to the Author” section, enter your conflict of interest statement in the “Confidential to Editor” section, and submit your "Accept" recommendation.

Reviewer #1: (No Response)

Reviewer #3: (No Response)

2. Is the manuscript technically sound, and do the data support the conclusions?

Reviewer #1: (No Response)

Reviewer #3: Partly

3. Has the statistical analysis been performed appropriately and rigorously? 

Reviewer #1: (No Response)

Reviewer #3: Yes

4. Have the authors made all data underlying the findings in their manuscript fully available?

Reviewer #1: (No Response)

Reviewer #3: Yes

5. Is the manuscript presented in an intelligible fashion and written in standard English?

Reviewer #1: (No Response)

Reviewer #3: Yes

6. Review Comments to the Author

Reviewer #1: The authors has done extensive revision to address the reviewers comments. I have only a minor suggestion to shorten the conclusion and convey the main contribution/message/conclusion of this paper.

Reviewer #3: The objective of this paper is to achieve automated diagnosis of lumbar spinal stenosis by measuring the anteroposterior diameter and foraminal widths of lumbar spine. The proposed framework is not novel because it only adopts SegNet and the contour evolution algorithm is a project solution rather than focusing on research innovation.

7. PLOS authors have the option to publish the peer review history of their article (what does this mean?). If published, this will include your full peer review and any attached files.

Reviewer #1: No

Reviewer #3: No

---

## [Author Response · Author response to Decision Letter 1]

9 Oct 2020

Response to the Reviewers

First of all, we would like to thank the reviewers for reviewing the latest version of our paper. Your comments and feedback are very valuable for improving the quality of the paper. We have made the minor but necessary change to our paper based on your feedback. Please find below our response to specific areas of the paper and the outline of the changes that we made. I hope that the change and our explanation are to your satisfaction.

Reviewer #1.

1. If the authors have adequately addressed your comments raised in a previous round of review and you feel that this manuscript is now acceptable for publication, you may indicate that here to bypass the “Comments to the Author” section, enter your conflict of interest statement in the “Confidential to Editor” section, and submit your "Accept" recommendation.

Reviewer #1: (No Response)

2. Is the manuscript technically sound, and do the data support the conclusions?

Reviewer #1: (No Response)

3. Has the statistical analysis been performed appropriately and rigorously? 

Reviewer #1: (No Response)

4. Have the authors made all data underlying the findings in their manuscript fully available?

Reviewer #1: (No Response)

5. Is the manuscript presented in an intelligible fashion and written in standard English?

Reviewer #1: (No Response)

6. Review Comments to the Author

Reviewer #1: The authors has done extensive revision to address the reviewers comments. I have only a minor suggestion to shorten the conclusion and convey the main contribution/message/conclusion of this paper.

Our response:

Thank you for confirming that our last revision has addressed all the reviewers’ comments. We have made the suggested minor change to the conclusion by making it more concise to emphasise the main message of the paper. Specifically, we removed the inclusion of specific results that were already provided in the abstract. 

The last four sentences in the conclusion that state: 

Our experiments show that our method can locate important points that are on region boundaries between 0.37 – 1.36 mm of the expert-located reference points. The average error of the calculated right and left foraminal distances relative to their expert-measured distances are 0.28 mm (p = 0.92) and 0.29 mm (p = 0.97), respectively. The average error of the calculated AP diameter relative to their expert-measured diameter is 0.90 mm (p = 0.92). The method also achieves 96.7% agreement with an expert opinion on determining the severity of the IVD herniation.

have been replaced with:

Our experiments show that our method can locate all the important points that are on region boundaries accurately. This allows accurate calculation of the AP diameter, which in turn, produces high agreement with the expert’s opinion when determining the severity of the IVD herniations.

Reviewer #3.

1. If the authors have adequately addressed your comments raised in a previous round of review and you feel that this manuscript is now acceptable for publication, you may indicate that here to bypass the “Comments to the Author” section, enter your conflict of interest statement in the “Confidential to Editor” section, and submit your "Accept" recommendation.

Reviewer #3: (No Response)

2. Is the manuscript technically sound, and do the data support the conclusions?

Reviewer #3: Partly

3. Has the statistical analysis been performed appropriately and rigorously? 

Reviewer #3: Yes

4. Have the authors made all data underlying the findings in their manuscript fully available?

Reviewer #3: Yes

5. Is the manuscript presented in an intelligible fashion and written in standard English?

Reviewer #3: Yes

6. Review Comments to the Author

Reviewer #3: The objective of this paper is to achieve automated diagnosis of lumbar spinal stenosis by measuring the anteroposterior diameter and foraminal widths of lumbar spine. The proposed framework is not novel because it only adopts SegNet and the contour evolution algorithm is a project solution rather than focusing on research innovation.

Our response:

We appreciate your comment regarding the novelty of our approach. Although application of SegNet and Contour Evolution algorithms themselves are not novel, we believe we have shown through our extensive set of experiments that our contour evolution technique can provide significant improvement relative to using SegNet alone or the benchmark contour evolution techniques. We believe that our approach could benefit other researchers when they are faced with similar problems. With that I hope our paper can be deemed to have sufficient merit and appropriate for publication.

---

## [Editor Report · Decision Letter 2]

13 Oct 2020

Automated measurement of anteroposterior diameter and foraminal widths in MRI images for lumbar spinal stenosis diagnosis

PONE-D-20-11502R2

Dear Dr. Sudirman,

We’re pleased to inform you that your manuscript has been judged scientifically suitable for publication and will be formally accepted for publication once it meets all outstanding technical requirements.

Kind regards,

Dzung Pham

Academic Editor

PLOS ONE
---

## [Editor Report · Acceptance letter]

23 Oct 2020

PONE-D-20-11502R2 

Automated measurement of anteroposterior diameter and foraminal widths in MRI images for lumbar spinal stenosis diagnosis 

Dear Dr. Sudirman:

I'm pleased to inform you that your manuscript has been deemed suitable for publication in PLOS ONE. Congratulations! Your manuscript is now with our production department. 

Kind regards, 

on behalf of

Dr Dzung Pham 

Academic Editor

PLOS ONE